# Simplifying Model-based RL: Learning Representations, Latent-space Models, and Policies with One Objective

**Raj Ghugare**[1]     **Homanga Bharadhwaj**[2]     **Benjamin Eysenbach**[2]
**Sergey Levine**[3]     **Ruslan Salakhutdinov**[2]

[1]VNIT Nagpur     [2]Carnegie Mellon University     [3]UC Berkeley

raj19@students.vnit.ac.in, hbharadh@cs.cmu.edu, beysenba@cs.cmu.edu

## Abstract

While reinforcement learning (RL) methods that learn an internal model of the environment have the potential to be more sample efficient than their model-free counterparts, learning to model raw observations from high dimensional sensors can be challenging. Prior work has addressed this challenge by learning low-dimensional representation of observations through auxiliary objectives, such as reconstruction or value prediction. However, the alignment between these auxiliary objectives and the RL objective is often unclear. In this work, we propose a single objective which jointly optimizes a latent-space model and policy to achieve high returns while remaining self-consistent. This objective is a lower bound on expected returns. Unlike prior bounds for model-based RL on policy exploration or model guarantees, our bound is directly on the overall RL objective. We demonstrate that the resulting algorithm matches or improves the sample-efficiency of the best prior model-based and model-free RL methods. While sample efficient methods typically are computationally demanding, our method attains the performance of SAC in about 50% less wall-clock time[1].

## 1 Introduction

While RL algorithms that learn an internal model of the world can learn more quickly than their model-free counterparts (Hafner et al., 2018; Janner et al., 2019), figuring out exactly *what* these models should predict has remained an open problem: the real world and even realistic simulators are too complex to model accurately. Although model errors may be rare under the training distribution, a learned RL agent will often seek out the states where an otherwise accurate model makes mistakes (Jafferjee et al., 2020). Simply training the model with maximum likelihood will not, in general, produce a model that is good for model-based RL (MBRL). The discrepancy between the policy objective and the model objective is called the objective mismatch problem (Lambert et al., 2020), and remains an active area of research. The objective mismatch problem is especially important in settings with high-dimensional observations, which are challenging to predict with high fidelity.

Prior model-based methods have coped with the difficulty to model high-dimensional observations by learning the dynamics of a compact representation of observations, rather than the dynamics of the raw observations. Depending on their learning objective, these representations might still be hard to predict or might not contain task relevant information. Besides, the accuracy of prediction depends not just on the model's parameters, but also on the states visited by the policy. Hence, another way of reducing prediction errors is to optimize the policy to avoid transitions where the model is inaccurate, while achieving high returns. In the end, we want to train the model, representations, and policy to be self-consistent: the policy should only visit states where the model is accurate, the representation should encode information that is task-relevant and predictable. *Can we design a model-based RL algorithm that automatically learns compact yet sufficient representations for model-based reasoning?*

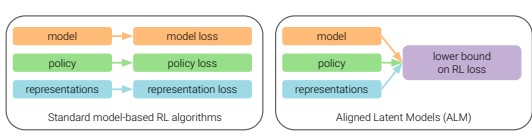

Figure 1: (*left*) Most model-based RL methods learn the representations, latent-space model, and policy using three different objectives. (*Right*) We derive a single objective for all three components, which is a lower bound on expected returns. Based on this objective, we develop a practical deep RL algorithm.

---

[1]Project website with code: https://alignedlatentmodels.github.io/

In this paper, we present a simple yet principled answer to this question by devising a single objective that *jointly* optimizes the three components of the model-based algorithm: the representation, the model, and the policy. As shown in Fig. 1, this is in contrast to prior methods, which use three separate objectives. We build upon prior work that views model-based RL as a latent-variable problem: the objective is to maximize the returns (the likelihood), which is an expectation over trajectories (the unobserved latent variable) (Botvinick & Toussaint, 2012; Attias, 2003; Eysenbach et al., 2021a). This is different from prior work that maximizes the likelihood of observed data, independent of the reward function (Hafner et al., 2019; Lee et al., 2020). This perspective suggests that model-based RL algorithms should resemble inference algorithms, sampling trajectories (the latent variable) and then maximizing returns (the likelihood) on those trajectories. However, sampling trajectories is challenging when observations are high-dimensional. The key to our work is to infer both the trajectories (observations, actions) and the representations of the observations. Crucially, we show how to maximize the expected returns under this inferred distribution by sampling only the representations, without the need to sample high-dimensional observations.

The main contribution of this paper is Aligned Latent Models (ALM), an MBRL algorithm that jointly optimizes the observation representations, a model that predicts those representations, and a policy that acts based on those representations. To the best of our knowledge, this objective is the first lower bound for a model-based RL method with a latent-space model. Across a range of continuous control tasks, we demonstrate that ALM achieves higher sample efficiency than prior model-based and model-free RL methods, including on tasks that stymie prior MBRL methods. Because ALM does not require ensembles (Chua et al., 2018; Janner et al., 2019) or decision-time planning (Deisenroth & Rasmussen, 2011; Sikchi et al., 2020; Morgan et al., 2021), our open-source implementation performs updates $10\times$ and $6\times$ faster than MBPO (Janner et al., 2019) and REDQ (Chen et al., 2021) respectively, and achieves near-optimal returns in about 50% less time than SAC.

## 2 RELATED WORK

Prior model-based RL methods use models in many ways, using it to search for optimal action sequences (Garcia et al., 1989; Springenberg et al., 2020; Hafner et al., 2018; Chua et al., 2018; Hafner et al., 2019; Xie et al., 2020), to generate synthetic data (Sutton, 1991; Luo et al., 2018; Hafner et al., 2019; Janner et al., 2019; Shen et al., 2020), to better estimate the value function (Deisenroth & Rasmussen, 2011; Chua et al., 2018; Buckman et al., 2018; Feinberg et al., 2018), or some combination thereof (Schrittwieser et al., 2020; Hamrick et al., 2020; Hansen et al., 2022). Similar to prior work on stochastic value gradients (Heess et al., 2015; Hafner et al., 2019; Clavera et al., 2020; Amos et al., 2020), our approach uses model rollouts to estimate the value function for a policy gradient. Prior work find that taking gradients through a learned dynamics model can be unstable (Metz et al., 2021; Parmas et al., 2019). However, similar to dreamer Hafner et al. (2019; 2020), we found that BPTT can work successfully if done with a latent-space model with appropriately regularized representations. Unlike these prior works, the precise form of the regularization emerges from our principled objective.

Because learning a model of high-dimensional observations is challenging, many prior model-based methods first learn a compact representation using a representation learning objective (e.g., image reconstruction (Kaiser et al., 2019; Oh et al., 2015; Buesing et al., 2018; Ha & Schmidhuber, 2018; Hafner et al., 2018; 2019; 2020), value and action prediction (Oh et al., 2017; Schrittwieser et al., 2020; Grimm et al., 2020), planning performance (Tamar et al., 2016; Racanière et al., 2017; Okada et al., 2017), or self-supervised learning (Deng et al., 2021; Nguyen et al., 2021; Okada & Taniguchi, 2020)). These methods then learn the dynamics of these representations (not of the raw observations), and use the model for RL. The success of these methods depends on the representation Arumugam & Roy (2022): the representations should be compact (i.e., easy to predict) while retaining task-relevant information. However, prior work does not optimize for this criterion, but instead optimizes the representation using some auxiliary objective.

The standard RL objective is to to maximize the expected returns, but models are typically learned via a different objective (maximum likelihood) and representations are learned via a third objective (e.g., image reconstruction). To solve this *objective mismatch* (Lambert et al., 2020; Joseph et al., 2013; Grimm et al., 2020), prior work study decision aware loss functions which optimize the model to minimize the difference between true and imagined next step values (Farahmand et al., 2017; Farahmand, 2018; D'Oro et al., 2020; Abachi et al., 2020; Voelcker et al., 2022) or directly optimize

the model to produce high-return policies (Eysenbach et al., 2021a; Amos et al., 2018; Nikishin et al., 2021). However, effectively addressing the objective mismatch problem for latent-space models remains an open problem. Our method makes progress on this problem by proposing a single objective to be used for jointly optimizing the model, policy, and representation. Because all components are optimized with the same objective, updates to the representations make the policy better (on this objective), as do updates to the model. While prior theoretical work in latent space models has proposed bounds on exploratory behavior of the policy (Misra et al., 2020), and on learning compressed latent representations (Efroni et al., 2021), our analysis lifts some of the assumptions (e.g., removing the block-MDP assumption), and bounds the overall RL objective in a model-based setting.

## 3 A UNIFIED OBJECTIVE FOR LATENT-SPACE MODEL-BASED RL

We first introduce notation, then provide a high-level outline of the objective, and then derive our objective. Sec. 4 will discuss a practical algorithm based on this objective.

### 3.1 PRELIMINARIES

The agent interacts with a Markov decision process (MDP) defined by states $s_t$, actions $a_t$, an initial state distribution $p_0(s)$, a dynamics function $p(s_{t+1} \mid s_t, a_t)$, a positive reward function $r(s_t, a_t) \geq 0$ and a discount factor $\gamma \in [0, 1)$. The RL objective is to learn a policy $\pi(a_t \mid s_t)$ that maximizes the discounted sum of expected rewards within an infinite-horizon episode:

$$\max_{\pi} \mathbb{E}_{s_{t+1} \sim p(\cdot|s_t,a_t), a_t \sim \pi(\cdot|s_t)} \left[ (1 - \gamma) \sum_{t=0}^{\infty} \gamma^t r(s_t, a_t) \right]. \tag{1}$$

The factor of $(1 - \gamma)$ does not change the optimal policy, but simplifies the analysis (Janner et al., 2020; Zahavy et al., 2021). We consider policies that are factored into two parts: an observation encoder $e_\phi(z_t \mid s_t)$ and a representation-conditioned policy $\pi_\phi(a_t \mid z_t)$. Our analysis considers infinite-length trajectories $\tau$, which include the actions $a_t$, observations $s_t$, and the corresponding observation representations $z_t$: $\tau \triangleq (s_0, a_0, z_0, s_1, a_1, z_1, \cdots)$. To simplify notation, we write the discounted sum of rewards as $R(\tau) \triangleq (1 - \gamma) \sum_{t=0}^{\infty} \gamma^t r(s_t, a_t)$. Lastly, we define the Q-function of a policy parameterized by $\phi$, as $Q(s_t, a_t) = \mathbb{E}_{\tau \sim \pi_\phi, e_\phi, p} [R(\tau) \mid s_t, a_t]$.

### 3.2 METHOD OVERVIEW

Our method consists of three components, shown in Fig. 2. The *first component* is an encoder $e_\phi(z_t \mid s_t)$, which takes as input a high-dimensional observation $s_t$ and produces a compact representation $z_t$. This representation should be as compact as possible, while retaining the bits for selecting good actions and for predicting the Q-function. The *second component* is a dynamics model of representations, $m_\phi(z_{t+1} \mid z_t, a_t)$, which takes as input the representation of the current observation and the action and predicts the representation of the next observation. The *third component* is a policy $\pi_\phi(a_t \mid z_t)$, which takes representations as inputs and chooses an action. This policy will be optimized to select actions to maximize rewards, while also keeping the agent in states where the dynamics model is accurate.

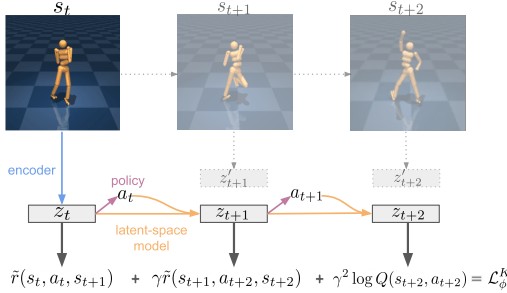

Figure 2: **Aligned Latent Models (ALM)** performs model-based RL by jointly optimizing the policy, the latent-space model, and the representations produced by the encoder using the same objective: maximize predicted rewards while minimizing the errors in the predicted representations. This objective corresponds to RL with an augmented reward function $\tilde{r}$. ALM estimates this objective without predicting high-dimensional observations $s_{t+1}$.

### 3.3 DERIVING THE OBJECTIVE

To derive our objective, we build on prior work (Toussaint, 2009; Kappen et al., 2012) and view the RL objective as a latent-variable problem, where the return $R(\tau)$ is the likelihood and the trajectory $\tau$

is the latent variable. Different from prior work, we include *representations* in this trajectory, in addition to the raw states, a difference which allows our method to learn good representations for MBRL. We can write the RL objective (Eq. 1) in terms of trajectories as $\mathbb{E}_{p(\tau)}[R(\tau)]$ by defining the distribution over trajectories $p(\tau)$ as

$$p_\phi(\tau) \triangleq p_0(s_0) \prod_{t=0}^{\infty} p(s_{t+1} \mid s_t, a_t)\pi_\phi(a_t \mid z_t)e_\phi(z_t \mid s_t). \tag{2}$$

Estimating and optimizing this objective directly is challenging because drawing samples from $p(\tau)$ requires interacting with the environment, an expensive operation. What we would like to do instead is estimate this same expectation via trajectories sampled from a different distribution, $q(\tau)$. We can estimate a lower bound on the expected return objective using samples from this different objective, by using the standard evidence lower bound (Jordan et al., 1999):

$$\log \mathbb{E}_{p(\tau)}[R(\tau)] \geq \mathbb{E}_{q(\tau)}\left[\log R(\tau) + \log p(\tau) - \log q(\tau)\right]. \tag{3}$$

This lower bound resolves a first problem, allowing us to estimate (a bound on) the expected return by drawing samples from the learned model, rather than from the true environment. However, learning a distribution over trajectories is difficult due to challenges in modeling high-dimensional observations, and potential compounding errors during sampling that can cause the policy to incorrectly predict high returns. We resolve these issues by only sampling compact representations of observations, instead of the observations themselves. By learning to predict the rewards and Q-values as a function of these representations, we are able to estimate this lower bound without sampling high-dimensional observations. Further, we carefully parameterize the learned distribution $q(\tau)$ to support an arbitrary length of model rollouts ($K$), which allows us to estimate the lower bound accurately:

$$q_\phi^K(\tau) = p_0(s_0)e_\phi(z_0 \mid s_0)\pi_\phi(a_0 \mid z_0) \prod_{t=1}^{K} p(s_t \mid s_{t-1}, a_{t-1})m_\phi(z_t \mid z_{t-1}, a_{t-1})\pi_\phi(a_t \mid z_t). \tag{4}$$

While it may seem strange that the future representations sampled from $q_\phi^K(\tau)$ are independent of states, this is an important design choice. It allows us to estimate the lower bound for any policy, using only samples from the latent-space model, without access to high dimensional states from the environment's dynamics function.

Combining the lower bound (Eq. 3) with this choice of parameterization, we obtain the following objective for model-based RL:

$$\mathcal{L}_\phi^K \triangleq \mathbb{E}_{q_\phi^K(\tau)}\left[\left(\sum_{t=0}^{K-1} \gamma^t \tilde{r}(s_t, a_t, s_{t+1})\right) + \gamma^K \log Q(s_K, a_K)\right], \tag{5}$$

$$\text{where} \quad \tilde{r}(s_t, a_t, s_{t+1}) = \underbrace{(1-\gamma)\log r(s_t, a_t)}_{(a)} + \underbrace{\log e_\phi(z_{t+1} \mid s_{t+1}) - \log m_\phi(z_{t+1} \mid z_t, a_t)}_{(b)}. \tag{6}$$

This objective is an evidence lower bound on the RL objective (see Proof in Appendix A.2).

**Theorem 3.1.** *For any representation $e_\phi(z_t \mid s_t)$, latent-space model $m_\phi(z_{t+1} \mid z_t, a_t)$, policy $\pi_\phi(a_t \mid z_t)$ and $K \in \mathbb{N}$, the ALM objective $\mathcal{L}_\phi^K$ corresponds to a lower bound on the expected return objective:*

$$\frac{1}{1-\gamma}\exp(\mathcal{L}_\phi^K) \leq \mathbb{E}_{p_\phi(\tau)}\left[\sum_t \gamma^t r(s_t, a_t)\right].$$

Here, we provide intuition for our objective and relate it to objectives in prior work. We start by looking at the augmented reward. The first term (a: extrinsic term) in this augmented reward function is the log of true rewards, which is analogous to maximizing the true reward function in the real environment, albeit on a different scale. The second term (b: intrinsic term), i.e., the negative KL divergence between the latent-space model and the encoder, is reminiscent of the prior methods (Goyal et al., 2019; Eysenbach et al., 2021b; Bharadhwaj et al., 2021; Rakelly et al., 2021) that regularize the encoder against a prior, to limit the number of bits used from the observations. Taken together, all the components are trained to make the model self-consistent with the policy and representation.

---

**Algorithm 1** The ALM objective can be optimized with any RL algorithm. We present an implementation based on DDPG (Lillicrap et al., 2015).

---

1: Initialize the encoder $e_\phi(z_t \mid s_t)$, model $m_\phi(z_{t+1} \mid z_t, a_t)$, policy $\pi_\phi(a_t \mid z_t)$, classifier $C_\theta(z_{t+1}, a_t, z_t)$, reward $r_\theta(z_t, a_t)$, Q-function $Q_\theta(z_t, a_t)$, replay buffer $\mathcal{B}$.
2: **for** $n = 1, \cdots, N$ do **do**
3:     Select action $a_n \sim \pi_\phi(\cdot \mid e_\phi(s_n))$ using the current policy.
4:     Execute action $a_n$ and observe reward $r_n$ and next state $s_{n+1}$.
5:     Store transition $(s_n, a_n, r_n, s_{n+1})$ in $\mathcal{B}$.
6:     Sample length-$K$ sequences from the replay buffer $(s_i, a_i, s_{i+1})_{i=t}^{t+K-1} \sim \mathcal{B}$
7:     Compute the objective $\mathcal{L}_{e_\phi, m_\phi}^K((s_i, a_i, s_{i+1})_{i=t}^{t+K-1})$ using the sampled sequence.     ▷ Eq. 7
8:     Update encoder and model by gradient ascent on $\mathcal{L}_{e_\phi, m_\phi}^K$.
9:     Compute the objective $\mathcal{L}_{\pi_\phi}^K((s_{i=t}))$ using on-policy model-based trajectories.     ▷ Eq. 8
10:     Update policy by gradient ascent on $\mathcal{L}_{\pi_\phi}^K$.
11:     Update classifier, Q-function and reward using losses $\mathcal{L}_{C_\theta}, \mathcal{L}_{Q_\theta}, \mathcal{L}_{r_\theta}$.     ▷ Eq. 9, 10, 11

---

The last part of our objective is the length of rollouts $(K)$, that the model is used for. Our objective is directly applicable to all prior model-based RL algorithms which use the model only for a fixed number of rollouts rather than the entire horizon, like SVG style updates (Amos et al., 2020; Heess et al., 2015) and trajectory optimization (Tedrake, 2022). Larger values of $K$ correspond to looser bounds (see Appendix A.6). Although this suggests that a model-free estimate is the tightest, a Q-function learned using function approximation with TD-estimates (Thrun & Schwartz, 1993) is biased and difficult to learn. A larger value of $K$ decreases this bias by reducing the dependency on a learned Q-function. While the lower bound in Theorem 3.1 is not tight, we can include a learnable discount factor such that it becomes tight (see Appendix A.5). In our experiments 4, we find that the objective in Theorem 3.1 is still a good estimate of true expected returns.

In Appendix A.4, we derive a closed loop form for the optimal latent-dynamics and show that they are biased towards high-return trajectories: they reweight the true probabilities of trajectories with their rewards. We also derive a lower bound for the model-based offline RL setting, obtaining a similar objective with an additional behavior cloning term (Appendix A.7).

## 4    A PRACTICAL ALGORITHM

We now describe a practical method to jointly train the policy, model, and encoder using the lower bound ($\mathcal{L}_\phi^K$). We call the resulting algorithm Aligned Latent Models (ALM) because joint optimization means that the objectives for the model, policy, and encoder are the same; they are aligned. For training the encoder and model (latent-space learning phase), $q_\phi^K(\tau)$ is unrolled using actions from a replay buffer, whereas for training the policy (planning phase), $q_\phi^K(\tau)$ is unrolled using actions imagined from the latest policy. To estimate our objective using just representations, our method also learns to predict the reward and Q-function from the learned representations $z_t$ using real data only (see Appendix C for details). Algorithm 1 provides pseudocode.

**Maximizing the objective with respect to the encoder and latent-space model.**   To train the encoder and the latent-space model, we estimate the objective $\mathcal{L}_\phi^K$ using K-length sequences of transitions $\{s_i, a_i, s_{i+1}\}_{i=t}^{t+K-1}$ sampled from the replay buffer:

$$\mathcal{L}_{e_\phi, m_\phi}^K(\{s_i, a_i, s_{i+1}\}_{i=t}^{t+K-1}) = \mathbb{E}_{\substack{e_\phi(z_{i=t} \mid s_t) \\ m_\phi(z_{i>t} \mid z_{t:i-1}, a_{i-1})}} \Big[ \gamma^K Q_\theta(z_K, \pi(z_K))$$
$$+ \sum_{i=t}^{t+K-1} \gamma^i \big( r_\theta(z_i, a_i) - \mathrm{KL}(m_\phi(z_{i+1} \mid z_i, a_i) \| e_{\phi_{\mathrm{targ}}}(z_{i+1} \mid s_{i+1})) \big) \Big]. \quad (7)$$

To optimize this objective, we sample an initial representation from the encoder and roll out the latent-space model using action sequences taken in the real environment (see Fig. 2)[2]. We find that using a target encoder $e_{\phi_{\mathrm{targ}}}(z_t \mid s_t)$ to calculate the KL consistency term leads to stable learning.

---

[2]In our code, we do not use the $\gamma$ discounting for Equation 7.

**Maximizing the objective with respect to the policy.** The latent-space model allows us to evaluate the objective for the current policy by generating on-policy trajectories. Starting from a sampled state $s_t$ the latent-space model is recurrently unrolled using actions from the current policy to generate a $K$-length trajectory of representations and actions $(z_{t:t+K}, a_{t:t+K})$. Calculating the intrinsic term in the augmented reward $(\log e_\phi(z_{t+1} \mid s_{t+1}) - \log m_\phi(z_{t+1} \mid z_t, a_t))$ for on-policy actions is challenging, as we do not have access to the next high dimensional state $s_{t+1}$. Following prior work (Eysenbach et al., 2020; Eysenbach et al., 2021a), we learn a classifier to differentiate between representations sampled from the encoder $e_\phi(z_{t+1} \mid s_{t+1})$ and the latent-space model $m_\phi(z_{t+1} \mid z_t, a_1)$ to estimate this term (see Appendix C for details).

We train the latent policy by recurrently backpropagating stochastic gradients (Heess et al., 2015; Amos et al., 2020; Hafner et al., 2019) of our objective evaluated on this trajectory:

$$\mathcal{L}_{\pi_\phi}^K(s_t) = \mathbb{E}_{q_\phi^K(z_{t:K}, a_{t:K} \mid s_t)} \left[ \sum_{i=t}^{t+K-1} \gamma^{i-t} \left( r_\theta(z_i, a_i) + c \cdot \log \frac{C_\theta(z_{i+1}, a_i, z_i)}{1 - C_\theta(z_{i+1}, a_i, z_i)} \right) + \gamma^K Q_\theta(z_K, \pi(z_K)) \right]. \tag{8}$$

During policy training (Eq. 8), estimating the intrinsic reward, $\log e_\phi(z_{t+1} \mid s_{t+1}) - \log m_\phi(z_{t+1} \mid z_t, a_t)$, is challenging because we do not have access to the next high dimensional state $(s_{t+1})$. Following prior work (Eysenbach et al., 2020; Eysenbach et al., 2021a), we note that a learned classifier between representations sampled from the encoder $e_\phi(z_{t+1} \mid s_{t+1})$ versus the latent-space model $m_\phi(z_{t+1} \mid z_t, a_1)$ can also be used to estimate the difference between log-likelihoods under them, which is exactly equal to the augmented reward function:

$$\log e_\phi(z_{t+1} \mid s_{t+1}) - \log m_\phi(z_{t+1} \mid z_t, a_t) \approx \log \frac{C_\theta(z_{t+1}, a_t, z_t)}{1 - C_\theta(z_{t+1}, a_t, z_t)}.$$

Here, $C_\theta(z_{t+1}, a_t, z_t) \in [0, 1]$ is a learned classifier's prediction of the probability that $z_{t+1}$ is sampled from the encoder conditioned on the next state after starting at $(z_t, a_t)$ pair. The classifier is trained via the standard cross entropy loss:

$$\mathcal{L}_{C_\theta}(z_t \sim e_\phi(\cdot \mid s_t), z_{t+1}, \widehat{z_{t+1}}) = \log(C_\theta(z_{t+1}, z_t, a_t)) + \log(1 - C_\theta(\widehat{z_{t+1}}, z_t, a_t)). \tag{9}$$

where $z_{t+1} \sim e_\phi(\cdot \mid s_{t+1})$ is the *real* next representation and $\widehat{z_{t+1}} \sim m_\phi(\cdot \mid z_t, a_t)$ is the *imagined* next representation, both from the same starting pair $(z_t, a_t)$.

**Differences between theory and experiments.** While our theory suggests a coefficient $c = 1$, we use $c = 0.1$ in our experiments because it slightly improves the results. We do provide an ablation which shows that ALM performs well across different values of c Figure 16. We also omit the log of the true rewards in both Eq. 8 and 7. We show that this change is equivalent to the first one for all practical purposes (see Appendix A.8.). Nevertheless, these changes mean that the objective we use in practice is not guaranteed to be a lower bound.

## 5 EXPERIMENTS

Our experiments focus on whether jointly optimizing the model, representation, and policy yields benefits relative to prior methods that use different objectives for different components. We use SAC-SVG (Amos et al., 2020) as the main baseline, as it structurally resembles our method but uses different objectives and architectures. While design choices like ensembling (MBPO, REDQ) are orthogonal to our paper's contribution, we nonetheless show that ALM achieves similar sample efficiency MBPO and REDQ without requiring ensembling; as a consequence, it achieves good performance in $\sim 6\times$ less wall-clock time. Additional experiments analyze the Q-values, ablate components of our method, and visualize the learned representations and model. All plots and tables show the mean and standard deviation across five random seeds. Where possible, we used hyperparameters from prior works; for example, our network dimensions were taken from Amos et al. (2020). Additional implementation details and hyperparameters are in Appendix D and a summary of successful and failed experiments are in Appendix G. We have released the code[3].

---

[3]https://alignedlatentmodels.github.io/

Table 1: On the model-based benchmark from Wang et al. (2019), ALM outperforms model-based and model-free methods on ⁴/₅ tasks, often by a wide margin. We report mean and std. dev. across 5 random seeds. We use T-Humanoid-v2 and T-Ant-v2 to refer to the respective truncated environments from Wang et al. (2019).

| | T-Humanoid-v2 | T-Ant-v2 | HalfCheetah-v2 | Walker2d-v2 | Hopper-v2 |
|---|---|---|---|---|---|
| ALM(3) | **5306** ± **437** | **4887** ± **1027** | **10789** ± **366** | **3006** ± **1183** | 2546 ± 1074 |
| SAC-SVG(2) (Amos et al., 2020) | 501 ± 37 | 4473 ± 893 | 8752 ± 1785 | 448 ± 1139 | **2852** ± **361** |
| SAC-SVG(3) | 472 ± 85 | 3833 ± 1418 | 9220 ± 1431 | 878 ± 1533 | 2024 ± 1981 |
| SVG(1) (Heess et al., 2015) | 811.8 ± 241 | 185 ± 141 | 336 ± 387 | 252 ± 48 | 435 ± 163 |
| SLBO (Luo et al., 2018) | 1377 ± 150 | 200 ±40 | 1097 ± 166 | 207 ± 108 | 805 ± 142 |
| TD3 (Fujimoto et al., 2018) | 147 ± 0.7 | 870 ± 283 | 3016 ± 969 | -516 ± 812 | 1817 ± 994 |
| SAC (Haarnoja et al., 2018) | 1470 ± 794 | 548 ± 146 | 3460 ± 1326 | 166 ± 1318 | 788 ± 977 |

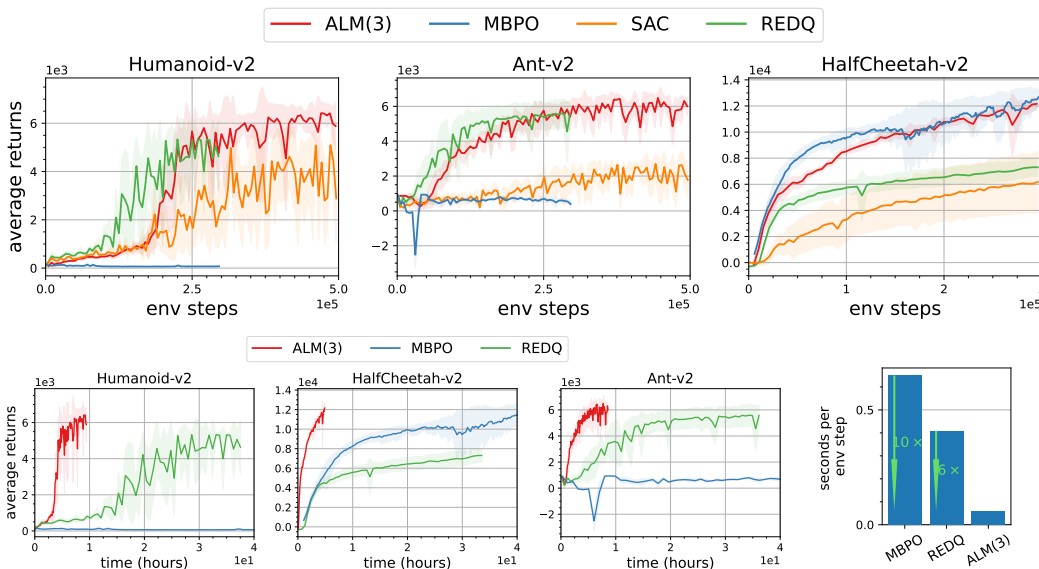

Figure 3: **Good performance without ensembles.** Our method (ALM) can (*Top*) match the sample complexity of ensembling-based methods (MBPO, REDQ) while (*Bottom*) requiring less runtime. Compared to MBPO, ALM takes ∼ 10× less time per environment step. See Appendix Fig. 9 for results on other environments.

**Baselines.** We provide a quick conceptual comparison to baselines in Table 2. In Sec. 5.1, we will compare with the most similar prior methods, SAC-SVG (Amos et al., 2020) and SVG (Heess et al., 2015). Like ALM, these methods use a learned model to perform SVG-style actor updates. SAC-SVG also maintains a hidden representation, using a GRU, to make recurrent predictions in the observation space. While SAC-SVG learns the model and representations using a reconstruction objective, ALM trains these components with the same objective as the policy. The dynamics model from SAC-SVG bears a resemblance to Dreamer-v2 (Hafner et al., 2020) and other prior work that targets image-based tasks (our experiments target state-based tasks). SAC-SVG reports better results than many prior model-based methods (POPLIN-P (Wang & Ba, 2019), SLBO (Luo et al., 2018), ME-TRPO (Kurutach et al., 2018)).

In Sec. 5.2, we focus on prior methods that use ensembles. MBPO (Janner et al., 2019) is a model-based method that uses an ensemble of dynamics models for both actor updates and critic updates. REDQ (Chen et al., 2021) is a model-free method which achieves sample efficiency on-par with model-based methods through the use of ensembles of Q-functions. We also compare to TD3 (Fujimoto et al., 2018) and SAC (Haarnoja et al., 2018); while typically not sample efficient, these methods can achieve good performance asymptotically.

## 5.1 IS THE ALM OBJECTIVE USEFUL?

We start by comparing ALM with the baselines on the locomotion benchmark proposed by Wang et al. (2019). In this benchmark, methods are evaluated based on the policy return after training for

2e5 environment steps. The TruncatedAnt-v2 and TruncatedHumanoid-v2 tasks included in this benchmark are easier than the standard Ant-v2 and Humanoid-v2 task, which prior model-based methods struggle to solve (Janner et al., 2019; Chua et al., 2018; Amos et al., 2020; Shen et al., 2020; Rajeswaran et al., 2020; Feinberg et al., 2018; Buckman et al., 2018). The results, shown in Table 1, show that ALM achieves better performance than the prior methods on 4/5 tasks. The results for SAC-SVG are taken from Amos et al. (2020) and the rest are from Wang et al. (2019). Because SAC-SVG is structurally similar to ALM, the better results from ALM highlight the importance of training the representations and the model using the same objective as the policy.

## 5.2   CAN ALM ACHIEVE GOOD PERFORMANCE WITHOUT ENSEMBLES?

Prior methods such as MBPO and REDQ use ensembles to achieve SOTA sample efficiency at the cost of long training times. We hypothesize that the self-consistency property of ALM will make the latent-dynamics simpler, allowing it to achieve good sample efficiency without the use of ensembles. Our next experiment studies whether ALM can achieve the benefits of ensembles without the computational costs. As shown in Figure 3, ALM matches the sample complexity of REDQ and MBPO, but requires $\sim 6\times$ less wall-clock time to train. Note that MBPO fails to solve the highest-dimensional tasks, Humanoid-v2 and Ant-v2 ($\mathbb{R}^{376}$ and $\mathbb{R}^{111}$). We optimized the ensemble training in the official REDQ code to be parallelized, leading to an increase in training speeds by $3\times$, which is still $2\times$ slower than our method (which does not employ parallelization optimization).

## 5.3   WHY DOES ALM WORK?

To better understand why ALM achieves high sample complexity without ensembles, we analyzed the Q-values, ran ablation experiments, and visualized the learned representations.

**Analyzing the Q-values.**   One way of inter-preting ALM is that it uses a model and an augmented reward function to obtain better estimates of the Q-values, which are used to train the policy. In contrast, REDQ uses a minimum of a random subset of the Q-function ensemble and SAC-AVG (baseline used in REDQ paper (Chen et al., 2021)) uses an average value of the Q-function ensemble to obtain a low variance estimate of these Q-values. While ensem-

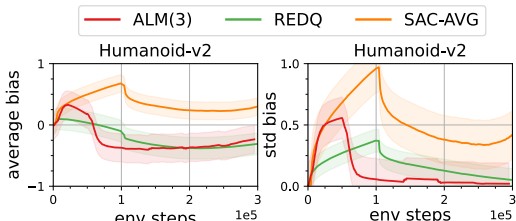

Figure 4: **Analyzing Q-values.** See text for details.

bles can be an effective way to improve the estimates of neural networks (Garipov et al., 2018; Abdar et al., 2021), we hypothesize that our latent-space model might be a more effective approach in the RL setting because it incorporates the dynamics, while also coming at a much lower computational cost.

To test this hypothesis, we measure the bias of the Q-values, as well as the standard deviation of that bias, following the protocol of Chen et al. (2021); Fujimoto et al. (2018). See Appendix E and Figure 4 for details. The positive bias will tell us whether the Q-values overestimates the true returns, while the standard deviation of this bias is more relevant for the purpose of selecting actions. We see from Fig. 4 that the standard deviation of the bias is lower for ALM than for REDQ and SAC-AVG, suggesting that the actions maximizing our objective are similar to actions that maximize true returns.

**Ablation experiments.**   In our first ablation experiment, we compare ALM to ablations that sepa-rately remove the KL term and the value term from the encoder objective (Eq. 7), and remove the classifier term from the policy objective (Eq. 8). As shown in Fig. 5a, the KL term, which is a purely self supervised objective Grill et al. (2020), is crucial for achieving good performance. The classifier term stabilizes learning (especially on Ant and Walker), while the value term has little effect. We hypothesize that the value term may not be necessary because its effect, driving exploration, may already be incorporated by Q-value overestimation (a common problem for RL algorithms Sutton & Barto (2018); Fujimoto et al. (2018)). A second ablation experiment (Fig. 5b) shows the performance of ALM for different numbers of unrolling steps ($K$). We perform a third ablation experiment of ALM(3), which uses the TD3 actor loss for training the policy. This ablation investigates whether the representations learned by ALM(3) are beneficial for model-free RL. Prior work (Gupta et al., 2017; Eysenbach et al., 2021b; Zhang et al., 2020) has shown that representations learning can

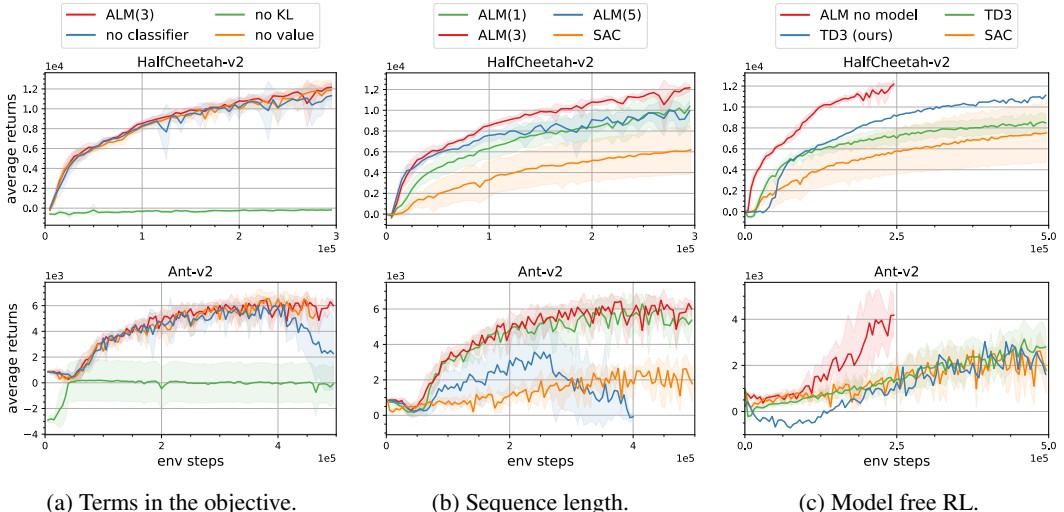

(a) Terms in the objective.  (b) Sequence length.  (c) Model free RL.

Figure 5: **Ablation experiments** (*Left*) Comparison of ALM (3) with no value term for the encoder, no KL term for the encoder and no classifier based rewards for the policy. Results reflect the importance of temporal consistency terms, especially for training the encoder. *(Center)* Comparison of ALM(K) for different values of K and baselines SAC. Using architectures that support larger values of $K$ could promise further improvements in performance. (*Right*) Representation learning objective of ALM(3) leads to higher sample efficiency for model-free RL. To ensure the validity of these results, we implemented TD3 (TD3 (ours)), which uses the same architecture, exploration and learning parameters as our method. Ablation results for other environments can be found in Fig. 10a, 10b, 10c.

facilitate properties like exploration, generalization and transfer. In fig. 5c, we find that the end to end representation learning of ALM(3) achieves high returns faster than standard model-free RL.

## 6 CONCLUSION

This paper introduced ALM, an objective for model-based RL that jointly optimizes representations, latent-space models, and policies all using the same objective. This objective mends the objective mismatch problem and results in a method where the representations, model, and policy all *cooperate* to maximize the expected returns. Our experiments demonstrate the benefits of such joint optimization: it achieves better performance than baselines that use separate objectives, and it achieves the benefits of ensembles without their computational costs.

At a high level, our end-to-end method is reminiscent of the success *deep* supervised learning. Deep learning methods promise to learn representations in an end-to-end fashion, allowing researchers to avoid manual feature design. Similarly, our algorithm suggests that the algorithmic components themselves, like representations and models, can be learned in an end-to-end fashion to optimize the desired objective.

**Limitations and future work.** The main limitation of our practical method is complexity: while simpler than prior model-based methods, it has more moving parts than model-free algorithms. One potential future work that can directly stem from ALM is an on-policy version of ALM, where Equation 8 can be calculated and simultaneously optimized with the encoder, model and policy, without using a classifier. Another limitation is that in our experiments, the value term in the equation 7, does not yield any benefit. An interesting direction is to investigate the reason behind this, or find tasks where an optimistic latent space is beneficial. While our theoretical analysis takes an important step in constructing lower bounds for model-based RL, it leaves many questions open, such as accounting for function approximation and exploration. Nonetheless, we believe that our proposed objective and method are not only practically useful, but may provide a template for designing even better model-based methods with learned representations.

**Acknowledgments.** The authors thank Shubham Tulsiani for helpful discussions throughout the project and feedback on the paper draft. We thank Melissa Ding, Jamie D Gregory, Midhun Sreekumar, Srikanth Vidapanakal, Ola Electric and CMU SCS for helping to set up the compute necessary for running the experiments. We thank Xinyue Chen and Brandon Amos for answering questions about the baselines used in the paper, and Nicklas Hansen for helpful discussions on model-based RL.

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

# A    PROOFS

## A.1    HELPER LEMMAS

**Lemma A.1.** *Let $P_K(H)$ be a truncated geometric distribution.*

$$P_K(H) = \begin{cases} (1-\gamma)\gamma^H & H \in [0, K-1] \\ \gamma^K & H = K \\ 0 & H > K \end{cases}$$

*Given discount factor $\gamma \in (0,1)$ and a random variable $x_t$, we have the following identity*

$$\mathbb{E}_{P_K(H)}\left[\sum_{t=0}^{H} x_t\right] = \sum_{H=0}^{K} P_K(H) \sum_{t=0}^{H} x_t$$

$$= (1-\gamma) \sum_{H=0}^{K-1} \gamma^t \sum_{t=0}^{H} x_t + \gamma^K \sum_{t=0}^{K} x_t$$

$$= x_0((1-\gamma)(1+\gamma+\cdots+\gamma^{K-1})+\gamma^K) + x_1((1-\gamma)(\gamma+\gamma^2+\cdots+\gamma^{K-1})+\gamma^K) + \cdots + x_K(\gamma^K)$$

$$= x_0((1-\gamma)\frac{1-\gamma^K}{1-\gamma}+\gamma^K) + x_1((1-\gamma)(\frac{\gamma(1-\gamma^{K-1})}{1-\gamma})+\gamma^K) + \cdots + x_K(\gamma^K)$$

$$= \sum_{t=0}^{K} \gamma^t x_t.$$

**Lemma A.2.** *Let $P_K(H)$ be a truncated geometric distribution and $p_\phi(\tau \mid H)$ be the distribution over $H+1$ length trajectories:*

$$p_\phi(\tau \mid H) = p_0(s_0)e_\phi(z_0 \mid s_0)\pi_\phi(a_0 \mid z_0)\prod_{t=1}^{H} p(s_t \mid s_{t-1}, a_{t-1})\pi_\phi(a_t \mid z_t)e_\phi(z_t \mid s_t)$$

*Then the RL objective can be re-written in the following way.*

$$\mathbb{E}_{p_\phi(\tau)}\left[(1-\gamma)\sum_{t}\gamma^t r(s_t, a_t)\right]$$

$$\mathbb{E}_{p_\phi^K(\tau)}\left[(1-\gamma)\sum_{t=0}^{K-1}\gamma^{t-1}r(s_t, a_t) + \gamma^K Q(s_K, a_K)\right]$$

$$= \mathbb{E}_{P_K(H)}\left[\mathbb{E}_{p_\phi(\tau|H=H)}\left[\mathbb{1}\{H \leq K-1\}r(s_H, a_H) + \mathbb{1}\{H=K\}Q(s_H, a_H)\right]\right].$$

The Q function is defined as $Q(s_H, a_H) = \mathbb{E}_{p_\phi(\tau)}[(1-\gamma)\sum_{t=0}^{\infty}\gamma^t r(s_{t+H}, a_{t+H})]$. This lemma helps us to interpret the discounting in RL as sampling from a truncated geometric distribution over future time steps.

**Lemma A.3.** *Let $p(x)$ be a distribution over $\mathbb{R}^n$. The following optimization problem can be solved using the methods of Lagrange multipliers to obtain an optimal solution analytically.*

$$\max_{p(x)} \mathbb{E}_{p(x)}\left[f(x) - \log p(x)\right]$$

$$such\ that \quad \int p(x)dx = 1$$

*Starting out by writing the Lagrangian for this problem, then differentiating it and then equating it to 0:*

$$\mathcal{L}(p(s), \lambda) \overset{a}{=} \mathbb{E}_{p(x)}[f(x) - \log p(x)] - \lambda(\int p(x)dx - 1)$$

$$\nabla_{p(x)}\mathcal{L} = f(x) - \log p(x) - 1 - \lambda$$

$$0 = f(x) - \log p(x) - 1 - \lambda$$

$$p^*(x) \overset{b}{=} e^{f(x) - 1 - \lambda}$$

We find the value of $\lambda$ by substituting the value of $p(x)$ from $(b)$ in the equality constraint.

$$\int e^{f(x) - 1 - \lambda}dx = 1$$

$$e^{1+\lambda} \overset{c}{=} \int e^{f(x)}dx$$

Substituting $(c)$ to remove the dual variable from $(b)$, we obtain

$$p^*(x) \overset{d}{=} \frac{e^{f(x)}}{\int e^{f(x)}dx}.$$

### A.2 A LOWER BOUND FOR K-STEP LATENT-SPACE

In this section we present the proof of Theorem 3.1. We restate the theorem for clarity.

**Theorem 3.1.** *For any representation $e_\phi(z_t \mid s_t)$, latent-space model $m_\phi(z_{t+1} \mid z_t, a_t)$, policy $\pi_\phi(a_t \mid z_t)$ and $K \in \mathbb{N}$, the ALM objective $\mathcal{L}_\phi^K$ corresponds to a lower bound on the expected return objective:*

$$\frac{1}{1-\gamma}\exp(\mathcal{L}_\phi^K) \leq \mathbb{E}_{p_\phi(\tau)}\left[\sum_t \gamma^t r(s_t, a_t)\right].$$

Note that scaling the rewards by a constant factor $(1-\gamma)$ does not change the RL problem and finding a policy to maximize the log of the expected returns is the same as finding a policy to maximize expected returns, because $\log$ is monotonic.

We want to estimate the RL objective with trajectories sampled from a different distribution $q_\phi(\tau)$ which leads to an algorithm that avoids sampling high-dimensional observations.

$$q_\phi(\tau) = p_0(s_0)e_\phi(z_0 \mid s_0)\prod_{t=0}^{\infty} p(s_{t+1} \mid s_t, a_t)m_\phi(z_{t+1} \mid z_t, a_t)\pi_\phi(a_t \mid z_t)$$

When used as trajectory generative models, $p_\phi(\tau)$ predicts representations $z_t$ using current state $s_t$. Whereas $q_\phi(\tau)$ predicts $z_t$ by unrolling the learned latent-model recurrently on $z_{t-1}$ (and $a_{t-1}$). Similar to variational inference, $p_\phi(\tau)$ can be interpreted as the posterior and $q_\phi(\tau)$ as the recurrent prior. Since longer recurrent predictions of a learned model significantly diverge from the true ones, we parameterize $q$ to support an arbitrary length of model rollouts $(K)$ during planning:

$$q_\phi^K(\tau) = p_0(s_0)e_\phi(z_0 \mid s_0)\pi_\phi(a_0 \mid z_0)\prod_{t=1}^{K} p(s_t \mid s_{t-1}, a_{t-1})m_\phi(z_t \mid z_{t-1}, a_{t-1})\pi_\phi(a_t \mid z_t)$$

*Proof.* We derive a lower bound on the RL objective for K-step latent rollouts.

$$\log \mathbb{E}_{p_\phi(\tau)} \left[ (1-\gamma) \sum_t \gamma^t r(s_t, a_t) \right]$$

$$\stackrel{\mathrm{a}}{=} \log \mathbb{E}_{p_\phi^K(\tau)} \left[ (1-\gamma) \sum_{t=0}^{K-1} \gamma^t r(s_t, a_t) + \gamma^K Q(s_K, a_K) \right]$$

$$\stackrel{\mathrm{b}}{=} \log \mathbb{E}_{P_K(H)} \left[ \mathbb{E}_{p_\phi(\tau|H=H)} \left[ \underbrace{\mathbb{1}\{H \leq K-1\} r(s_H, a_H) + \mathbb{1}\{H = K\} Q(s_H, a_H)}_{\Psi} \right] \right]$$

$$\stackrel{\mathrm{c}}{=} \log \iint P_K(H) p_\phi(\tau \mid H = H) \left( \Psi \right) d\tau dH$$

$$\stackrel{\mathrm{d}}{\geq} \int P_K(H) \log \int q_\phi(\tau \mid H = H) \frac{p_\phi(\tau \mid H = H)}{q_\phi(\tau \mid H = H)} \left( \Psi \right) d\tau dH$$

$$\stackrel{\mathrm{e}}{\geq} \iint P_K(H) q_\phi(\tau \mid H = H) \log \left( \frac{p_\phi(\tau \mid H = H)}{q_\phi(\tau \mid H = H)} \left( \Psi \right) \right) d\tau dH$$

$$\stackrel{\mathrm{f}}{=} \iint P_K(H) q_\phi(\tau \mid H = \infty) \left( \left( \sum_{t=0}^{H} \log e_\phi(z_{t+1} \mid s_{t+1}) - \log m_\phi(z_{t+1} \mid z_t, a_t) \right) + \log \left( \Psi \right) \right) d\tau dH$$

$$\stackrel{\mathrm{g}}{=} \iint q_\phi(\tau) P_K(H) \left( \left( \sum_{t=0}^{H} \log e_\phi(z_{t+1} \mid s_{t+1}) - \log m_\phi(z_{t+1} \mid z_t, a_t) \right) + \log \left( \Psi \right) \right) d\tau dH$$

$$\stackrel{\mathrm{h}}{=} \int q_\phi(\tau) \sum_{t=0}^{K-1} \gamma^t \left( \log e_\phi(z_{t+1} \mid s_{t+1}) - \log m_\phi(z_{t+1} \mid z_t, a_t) \right) + (1-\gamma)\gamma^t \log r(s_t, a_t) + \gamma^K \log Q(s_K, a_K) d\tau$$

$$\stackrel{\mathrm{i}}{=} \mathbb{E}_{q_\phi^K(\tau)} \left[ \sum_{t=0}^{K-1} \gamma^t \left( \log e_\phi(z_{t+1} \mid s_{t+1}) - \log m_\phi(z_{t+1} \mid z_t, a_t) \right) + (1-\gamma)\gamma^t \log r(s_t, a_t) + \gamma^K \log Q(s_K, a_K) \right]$$

$$\square$$

In $(a)$, we start with the $K$-step version of the RL objective. For $(b)$, we use Lemma A.2. For $(d)$, we use Jensen's inequality and multiply by $\frac{q_\phi(\tau|H=H)}{q_\phi(\tau|H=H)}$. For $(e)$, we apply Jensen's inequality. For $(f)$, since all the terms inside the summation only depends on the first H steps of the trajectory, we change the integration from over $H$ length to over infinite length trajectories. For $(g)$, we change the order of integration. For $(h)$, we use Lemma A.1 and the fact that $\mathbb{E}_{P_K(H)} [\log(\mathbb{1}\{H \leq K-1\} r(s_H, a_H) + \mathbb{1}\{H = K\} Q(s_H, a_H))] = \sum_{t=0}^{K-1} (1 - \gamma)\gamma^t \log r(s_t, a_t) + \gamma^K \log Q(z_K, a_K)$.

## A.3 A LOWER BOUND FOR LATENT-SPACE LAMBDA WEIGHTED SVG(K)

Using on-policy the data, the policy returns can be estimated using the k-step RL estimator ($\mathrm{RL}^k$) which uses the sum of rewards for the first k steps and truncates it with the Q-function.

$$\mathrm{RL}^k = (1-\gamma) \sum_{t=0}^{k-1} \gamma^t r(s_t, a_t) + \gamma^k Q(s_k, a_k)$$

Like previous works (Schulman et al., 2015; Hafner et al., 2019), we write the RL objective as a $\lambda$-weighted average of k-step returns for different horizons (different values of k), to substantially reduce the variance of the estimated returns at the cost of some bias.

$$\mathbb{E}_{p_\phi(\tau)} \left[ (1-\gamma) \sum_t \gamma^t r(s_t, a_t) \right] = \mathbb{E}_{p_\phi(\tau)} \left[ (1-\lambda) \sum_{k=1}^{\infty} \lambda^{n-1} \mathrm{RL}^k \right]$$

Our K-step lower bound $\mathcal{L}_\phi^K$ involves rolling out the model on-policy for K time-steps. A larger value of K reduces the dependency on a learned Q-function and hence reduces the estimation bias at the cost of higher variance.

**Lemma A.4.** *We show that a $\lambda$-weighted average of our lower bounds for different horizons (different values of K) is a lower bound on the RL objective.*

$$\log \mathbb{E}_{p_\phi(\tau)} \left[ (1 - \gamma) \sum_t \gamma^t r(s_t, a_t) \right]$$

$$\overset{a}{=} \log \mathbb{E}_{p_\phi(\tau)} \left[ (1 - \lambda) \sum_{k=1}^{\infty} \lambda^{k-1} RL^k \right]$$

$$\overset{b}{=} \log (1 - \lambda) \sum_{k=1}^{\infty} \lambda^{k-1} \mathbb{E}_{p_\phi(\tau)} \left[ RL^k \right]$$

$$\overset{c}{=} \log \mathbb{E}_{P_{Geom}(k)} \left[ \mathbb{E}_{p_\phi(\tau)} \left[ RL^{k+1} \right] \right]$$

$$\overset{d}{\geq} \mathbb{E}_{P_{Geom}(k)} \log \left[ \mathbb{E}_{p_\phi(\tau)} \left[ RL^{k+1} \right] \right]$$

$$\overset{e}{=} (1 - \lambda) \sum_{k=1}^{\infty} \lambda^{k-1} \log \left[ \mathbb{E}_{p_\phi(\tau)} \left[ RL^k \right] \right]$$

$$\overset{f}{\geq} (1 - \lambda) \sum_{k=1}^{\infty} \lambda^{k-1} \mathcal{L}^k(\phi)$$

In $(a)$ we use $\lambda$-weighted average estimation of the RL objective. In $(b)$ we use linearity of expectation. In $(c)$, we note that for every $k$, the coefficient of $RL^k$ is actually the probability of $k - 1$ under the geometric distribution. Hence, we rewrite it as an expectation over the geometric distribution. In $(d)$, we use Jensen's inequality. In $(e)$ we write out the probability values of the geometric distribution. In $(f)$, we use Theorem 3.1 for every value of $k$.

## A.4 Deriving the optimal latent dynamics and the optimal discount distribution

We define $\gamma_{\phi,K}(H)$ to be a learned discount distribution over the first $K + 1$ timesteps $\{0, 1, \ldots, K\}$. We use this learned discount factor when using data from imaginary rollouts. We start by lower bounding the RL objective and derive the optimal discount distribution $\gamma_{\phi,K}^*(H)$ and the optimal latent dynamics distribution $q^*(\tau \mid H)$ that maximizes this lower bound.

$$\overset{a}{=} \log \mathbb{E}_{P_K(H)} \left[ \mathbb{E}_{p_\phi(\tau|H=H)} \left[ \underbrace{\mathbb{1}\{H \leq K - 1\} r(s_H, a_H) + \mathbb{1}\{H = K\} Q(s_H, a_H)}_{\Psi} \right] \right]$$

$$\overset{b}{=} \log \iint \left( \frac{\gamma_{\phi,K}(H) q_\phi(\tau \mid H)}{\gamma_{\phi,K}(H) q_\phi(\tau \mid H)} P_K(H) p_\phi(\tau \mid H) \psi \right) d\tau dH$$

$$\overset{c}{\geq} \int \gamma_{\phi,K}(H) \int q_\phi(\tau \mid H) \log \frac{P_K(H) p_\phi(\tau \mid H) \psi}{\gamma_{\phi,K}(H) q_\phi(\tau \mid H)} d\tau dH$$

Given a horizon $H \in \{1, \cdots, K\}$, the optimal dynamics $q^*(\tau \mid H)$ can be calculated analytically:

$$q^*(\tau \mid H) \overset{d}{=} \frac{p_\phi(\tau \mid H) \psi}{\int p_\phi(\tau' \mid H) \psi d\tau'}$$

We substitute this value of $q^*$ in equation $(c)$:

$$\stackrel{e}{=} \int \gamma_{\phi,K}(H) \int \frac{p_\phi(\tau \mid H)\psi}{\int p_\phi(\tau' \mid H)\psi d\tau'} \log \frac{P_K(H)p_\phi(\tau \mid H)\psi \int p_\phi(\tau' \mid H)\psi d\tau'}{\gamma_{\phi,K}(H)p_\phi(\tau \mid H)\psi} d\tau dH$$

$$\stackrel{f}{=} \int \gamma_{\phi,K}(H) \log \left( \frac{P_K(H)}{\gamma_{\phi,K}(H)} \int p_\phi(\tau' \mid H)\psi d\tau' \right) dH$$

We now calculate the optimal discount distribution analytically $\gamma_{\phi,K}^*(H)$:

$$\gamma_{\phi,K}^*(H) \stackrel{g}{=} \frac{P_K(H) \int p_\phi(\tau' \mid H)\psi d\tau'}{\sum_{H=0}^{K} P_K(H) \int p_\phi(\tau' \mid H)\psi d\tau'}$$

In $(a)$, we rewrite the RL objective using Lemma A.2. In $(b)$, we multiply by $\frac{\gamma_{\phi,K}(H)q_\phi(\tau|H)}{\gamma_{\phi,K}(H)q_\phi(\tau|H)}$. In $(c)$, we use Jensen's inequality. In $(d)$ and $(g)$ we use the method of Lagrange multipliers to derive optimal distributions. We use Lemma A.3 for this result. Below we write the optimal latent-space in terms of rewards and Q-function.

**Writing out the optimal latent-space distribution**: The optimal latent-dynamics are non-Markovian and do not match MDP dynamics, but are optimistic towards high-return trajectories.

$$q^*(\tau \mid H) = \begin{cases} \frac{p_\phi(\tau|H)r(s_H,a_H)}{\mathbb{E}_{p_\phi}[r(s_H',a_H')]} & H \in [1, K-1] \\ \frac{p_\phi(\tau|H)Q(s_H,a_H)}{\mathbb{E}_{p_\phi}[Q(s_H',a_H')]} & H = K \end{cases}$$

**Writing out the optimal discount distribution**:

$$\gamma_{\phi,K}^*(H) = \begin{cases} \frac{(1-\gamma)\gamma^H \mathbb{E}_{p_\phi}[r(s_H,a_H)]}{\mathbb{E}_{p_\phi}[Q(s_0',a_0')]} & H \in [0, K-1] \\ \frac{\gamma^K \mathbb{E}_{p_\phi}[Q(s_H,a_H)]}{\mathbb{E}_{p_\phi}[Q(s_0',a_0')]} & H = K \\ 0 & H > K \end{cases}$$

### A.5 TIGHTENING THE K-STEP LOWER BOUND USING THE OPTIMAL DISCOUNT DISTRIBUTION AND THE OPTIMAL LATENT DYNAMICS

We start from equation $(f)$ from Appendix A.4 the previous proof which is a lower bound on the RL objective. To derive $(f)$, we have already substituted the optimal latent-dynamics. We now substitute the value of $\gamma_{\phi,K}^*(H)$ and verify that the lower bound $(f)$ leads to the original objective, suggesting that the bound is tight.

$$\stackrel{f}{=} \int \gamma_{\phi,K}(H) \log \left( \frac{P_K(H)}{\gamma_{\phi,K}(H)} \int p_\phi(\tau' \mid H)\psi d\tau' \right) dH$$

$$\stackrel{h}{=} \int \frac{P_K(H) \int p_\phi(\tau' \mid H)\psi d\tau'}{\sum_{H=0}^{K} P_K(H) \int p_\phi(\tau' \mid H)\psi d\tau'} \log \left( \frac{\cancel{P_K(H)} \sum_{H=0}^{K} P_K(H) \int p_\phi(\tau' \mid H)\psi d\tau'}{\cancel{P_K(H)} \cancel{\int p_\phi(\tau'\mid H)\psi d\tau'}} \cancel{\int p_\phi(\tau'\mid H)\psi d\tau'} \right) dH$$

$$\stackrel{i}{=} \frac{\log(\sum_{H=0}^{K} P_K(H) \int p_\phi(\tau' \mid H)\psi d\tau')}{\cancel{\sum_{H=0}^{K} P_K(H) \int p_\phi(\tau' \mid H)\psi d\tau'}} \int P_K(H) \cancel{\int p_\phi(\tau'\mid H)\psi d\tau'} dH$$

$$\stackrel{k}{=} \log \mathbb{E}_{p_\phi(\tau)} \left[ (1-\gamma) \sum_t \gamma^t r(s_t, a_t) \right]$$

In $(h)$ we substitute the value of $\gamma_{\phi,K}^*(H)$ and cancel out common terms. Since the integration in $(i)$ is over H, we bring out all the terms that do not depend on H. For $(k)$, we use the result from Lemma A.2.

Hence we have proved that the bound becomes tight when using the optimal discount and trajectory distributions.

## A.6 TIGHTNESS OF LOWER BOUND WITH LENGTH OF ROLLOUTS K

*Proof.* Proving $\mathcal{L}^K \geq \mathcal{L}^{K+1}$ for all $K \in \mathbb{N}$ will do.

$$
\begin{aligned}
&= \mathcal{L}^K \\
&= \mathbb{E}_{q_\phi^K(\tau)} \left[ \sum_{t=0}^{K-1} \gamma^t \underbrace{(\log e_\phi(z_{t+1} \mid s_{t+1}) - \log m_\phi(z_{t+1} \mid z_t, a_t) + (1-\gamma) \log r(s_t, a_t))}_{\tilde{r}(s_t, a_t, s_{t+1})} + \gamma^K \log Q(s_K, a_K) \right] \\
&= \mathbb{E}_{q_\phi^K(\tau)} \left[ \sum_{t=0}^{K-1} \gamma^t \tilde{r}(s_t, a_t, s_{t+1}) + \gamma^K \log E_{p_\phi(\tau' \mid s_0 = s_K, a_0 = a_K)} \left[ (1-\gamma) R(\tau') \right] \right] \\
&\geq \mathbb{E}_{q_\phi^K(\tau)} \left[ \sum_{t=0}^{K-1} \gamma^t \tilde{r}(s_t, a_t, s_{t+1}) + \gamma^K E_{q_\phi(\tau' \mid s_0 = s_K, a_0 = a_K)} \left[ \tilde{r}(s_K, a_K, s_{K+1}) + \gamma^{K+1} \log Q(s_{K+1}, a_{K+1}) \right] \right] \\
&= \mathbb{E}_{q_\phi^{K+1}(\tau)} \left[ \sum_{t=0}^{K} \gamma^t \tilde{r}(s_t, a_t, s_{t+1}) + \gamma^{K+1} \log Q(s_{K+1}, a_{K+1}) \right] = \mathcal{L}^{K+1}
\end{aligned}
$$

$\square$

## A.7 A LOWER BOUND FOR LATENT-SPACE SVG(K) IN OFFLINE RL

In the offline RL setting (Levine et al., 2020; Prudencio et al., 2022), we have access to a static dataset of trajectories from the environment. These trajectories are collected from one or more unknown policies. We derive a lower bound similar to Theorem 3.1. The main difference is that in Theorem 3.1, once a policy was updated ($\phi_t \to \phi_{t+1}$), we were able to collect new data using it, while in offline RL we have to re-use the same static data. Similar to Theorem 3.1, we define the distribution over trajectories $p(\tau)$:

$$
p_{\phi, \mathrm{b}}(\tau) \triangleq p_0(s_0) \prod_{t=0}^{\infty} p(s_{t+1} \mid s_t, a_t) \pi_{\mathrm{b}}(a_t \mid s_t) e_\phi(z_t \mid s_t),
$$

such that the offline dataset consists of trajectories sampled from this true distribution. We do not assume access to the data collection policies. Rather, $\pi_{\mathrm{b}}(a_t \mid s_t)$ is the behavior cloning policy obtained from the offline dataset. We want to estimate the RL objective with trajectories sampled from a different distribution $q_\phi(\tau)$:

$$
q_\phi(\tau) = p_0(s_0) e_\phi(z_0 \mid s_0) \prod_{t=0}^{\infty} p(s_{t+1} \mid s_t, a_t) m_\phi(z_{t+1} \mid z_t, a_t) \pi_\phi(a_t \mid z_t),
$$

which leads to an algorithm that avoids sampling high-dimensional observations.

Similar to Theorem 3.1, we want to find an encoder $e_\phi(z_t \mid s_t)$, a policy $\pi_\phi(a_t \mid s_t)$, and a model $m_\phi(z_{t+1} \mid z_t, a_t)$ to maximize the RL objective:

$$\max_\phi \log \mathbb{E}_{p_{\phi,b}(\tau)} \left[ (1-\gamma) \sum_t \gamma^t r(s_t, a_t) \right]$$

$$\overset{a}{=} \log \mathbb{E}_{p_{\phi,b}^K(\tau)} \left[ (1-\gamma) \sum_{t=0}^{K-1} \gamma^t r(s_t, a_t) + \gamma^K Q(s_t, a_t) \right]$$

$$\overset{c}{=} \log \iint P_K(H) p_{\phi,b}(\tau \mid H = H) \left( \Psi \right) d\tau dH$$

$$\overset{d}{\geq} \int P_K(H) \log \int q_\phi(\tau \mid H = H) \frac{p_{\phi,b}(\tau \mid H = H)}{q_\phi(\tau \mid H = H)} \left( \Psi \right) d\tau dH$$

$$\overset{e}{\geq} \iint P_K(H) q_\phi(\tau \mid H = H) \log \left( \frac{p_{\phi,b}(\tau \mid H = H)}{q_\phi(\tau \mid H = H)} \left( \Psi \right) \right) d\tau dH$$

$$\overset{f}{=} \iint P_K(H) q_\phi(\tau \mid H = \infty) \left( \left( \sum_{t=0}^H \log(\frac{e_\phi(z_{t+1} \mid s_{t+1}) \pi_b(a_{t+1} \mid s_{t+1})}{m_\phi(z_{t+1} \mid z_t, a_t) \pi_\phi(a_{t+1} \mid z_{t+1})}) \right) + \log \left( \Psi \right) \right) d\tau dH$$

$$\overset{g}{=} \iint q_\phi(\tau) P_K(H) \left( \left( \sum_{t=0}^H \log(\frac{e_\phi(z_{t+1} \mid s_{t+1}) \pi_b(a_{t+1} \mid s_{t+1})}{m_\phi(z_{t+1} \mid z_t, a_t) \pi_\phi(a_{t+1} \mid z_{t+1})}) \right) + \log \left( \Psi \right) \right) d\tau dH$$

$$\overset{h}{=} \int q_\phi(\tau) \sum_{t=0}^{K-1} \gamma^t \left( \log(\frac{e_\phi(z_{t+1} \mid s_{t+1}) \pi_b(a_{t+1} \mid s_{t+1})}{m_\phi(z_{t+1} \mid z_t, a_t) \pi_\phi(a_{t+1} \mid z_{t+1})}) \right) + (1-\gamma)\gamma^t \log r(s_t, a_t) + \gamma^K \log Q(s_K, a_K) d\tau$$

$$\overset{i}{=} \mathbb{E}_{q_\phi^K(\tau)} \left[ \sum_{t=0}^{K-1} \gamma^t \underbrace{\log(\frac{e_\phi(z_{t+1} \mid s_{t+1})}{m_\phi(z_{t+1} \mid z_t, a_t)})}_{\text{KL consistency term}} + \gamma^t \underbrace{\log(\frac{\pi_b(a_{t+1} \mid s_{t+1})}{\pi_\phi(a_{t+1} \mid z_{t+1})})}_{\text{Behaviour cloning term}} + (1-\gamma)\gamma^t \log r(s_t, a_t) + \gamma^K \log Q(s_K, a_K) \right]$$

The only main difference is between this proof and the proof A.2 of Theorem3.1 is in step $(f)$, where canceling out the common terms of $p(\tau)$ and $q(\tau)$ leaves an additional *behavior cloning* term. This derivation theoretically backs the additional behavior cloning term used in prior representation learning methods (Abdolmaleki et al., 2018; Peters et al., 2010) for the offline RL setting.

## A.8 OMISSION OF THE LOG OF REWARDS AND Q FUNCTION IS EQUIVALENT TO SCALING KL IN EQUATION 8

While our main results omit the logarithmic transformation of the rewards and Q function, in this section we describe that this omission is approximately equivalent to scaling the KL coefficient in Eq. 8.

Using these insights, we applied ALM, with the log of rewards, to a transformed MDP with a shifted reward function. A large enough constant, a, was added to all original rewards(which doesn't change the optimal policy assuming that there are no terminal states)

$$r_{\text{new}} = r + a.$$

Taking a log of this new reward is equal to the reward of the original objective, scaled by the constant, $a$

$$a(\log(r+a) - \log(a)) \approx r.$$

The additive term can be ignored because it won't contribute to the optimization. We plot both $y = r$ and $y = a(\log(r+a) - \log(a))$, to show that they are very similar for commonly used values of rewards in Figure 6. The scaling constant $a$ can be interpreted as the weight of the log of rewards,

relative to the KL term in the ALM objective. Hence changing this value is approximately equivalent to scaling the KL coefficient. The results, shown in Fig 12, show that this version of ALM which includes the logarithm transformation performs at par with the version of ALM without the logarithm. This results shows that we can add back the logarithm to ALM without hurting performance. For both Figure 6 and Figure 12, we used the value of $a = 10000$.

**ALM objective using the transformed reward**

$$\stackrel{a}{=} \mathbb{E}_{q_\phi^K(\tau)} \left[ \sum_{t=0}^{K-1} \gamma^t \left( \log e_\phi(z_{t+1} \mid s_{t+1}) - \log m_\phi(z_{t+1} \mid z_t, a_t) \right) + (1-\gamma)\gamma^t \log(r_{\text{new}}(s_t, a_t)) + \gamma^K \log Q_{\text{new}}(s_K, a_K) \right]$$

$$\stackrel{b}{=} \mathbb{E}_{q_\phi^K(\tau)} \left[ \left[ \sum_{t=0}^{K-1} \gamma^t \left( \log e_\phi(z_{t+1} \mid s_{t+1}) - \log m_\phi(z_{t+1} \mid z_t, a_t) \right) + (1-\gamma)\gamma^t \left( \frac{r(s_t, a_t)}{a} + \log(a) \right) \right. \right.$$
$$\left. \left. + \gamma^K \left( \frac{Q(s_K, a_K)}{a} + \log(a) \right) \right] \right]$$

$$\stackrel{c}{=} \mathbb{E}_{q_\phi^K(\tau)} \left[ \left[ \sum_{t=0}^{K-1} \gamma^t \left( \log e_\phi(z_{t+1} \mid s_{t+1}) - \log m_\phi(z_{t+1} \mid z_t, a_t) \right) + (1-\gamma)\gamma^t \left( \frac{r(s_t, a_t)}{a} \right) + \gamma^K \left( \frac{Q(s_K, a_K)}{a} \right) \right] \right]$$

$$\stackrel{d}{=} \mathbb{E}_{q_\phi^K(\tau)} \left[ \left[ \sum_{t=0}^{K-1} \gamma^t a \left( \log e_\phi(z_{t+1} \mid s_{t+1}) - \log m_\phi(z_{t+1} \mid z_t, a_t) \right) + (1-\gamma)\gamma^t r(s_t, a_t) + \gamma^K Q(s_K, a_K) \right] \right]$$

In $(a)$, we write the ALM objective using the new reward function. In $(b)$, we use the fact that $\log(r_{\text{new}}(s_t, a_t)) \approx \left( \frac{r(s_t, a_t)}{a} + \log(a) \right)$, and $\log Q_{\text{new}}(s_K, a_K) \equiv \left( \frac{Q(s_K, a_K)}{a} + \log(a) \right)$ for a large enough constant a. (See explanation above). In $(c)$, we remove the extra constants which add up to 0. In $(d)$, we note that scaling the rewards and Q function by $1/a$, is equivalent to scaling the KL term by $a$, which is a large enough constant.

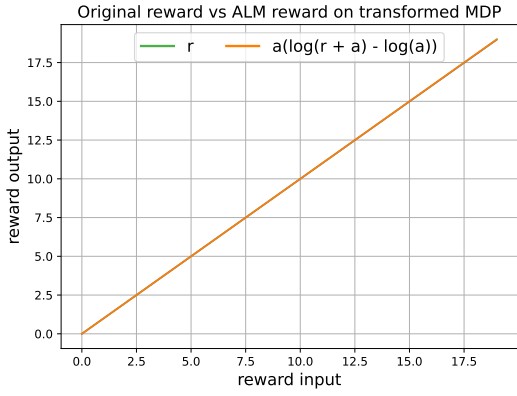

Figure 6: **Analyzing logarithmic reward transformation.**: While our theoretical derivation motivates a logarithmic transformation of the returns, here we show that appropriately scaling and shifting the rewards makes that logarithmic transformation have no effect. This throws light on the fact that omitting the logarithmic transformation can be thought of as scaling the two components in our augmented reward function Eq.6.

## B  COMPARISON TO BASELINES

Here we provide a quick comparison to baselines, conceptually in terms of number of gradient updates per env step, usage of ensembles are used for dynamics model or Q-function, and objectives for representations, model, and policy.

Table 2: Table showing conceptual comparisons of ALM to baselines.

|  | Ensembles | Representations | Model | Policy | UTD |
|---|---|---|---|---|---|
| ALM | ✗ | Lower bound | Lower bound | Lower bound | 3 |
| SAC-SVG | ✗ | MLE | MLE | Actor-Critic (stochastic value gradients) | 4 |
| MBPO | ✓ | ✗ | MLE | Soft Actor-Critic (dyna-style) | 20-40 |
| REDQ | ✓ | ✗ | ✗ | Soft Actor-Critic | 20 |

## C   ADDITIONAL LEARNING DETAILS

**Estimating the Q-function and reward function.**    Unlike most MBRL algorithms, the Q function $Q_\theta(z_t, a_t)$ is learned using transitions $s_t, a_t, r_t, s_{t+1}$ from the *real environment* only, using the standard TD loss:

$$\mathcal{L}_{Q_\theta}(z_t \sim e_\phi(\cdot \mid s_t), a_t, r_t, z_{t+1} \sim e_\phi(\cdot \mid s_{t+1})) = (Q_\theta(z_t, a_t) - (r_t + \gamma Q_{\theta_{\text{targ}}}(z_{t+1}, \pi(z_{t+1}))))^2. \tag{10}$$

The TD target is computed using a target Q-function (Fujimoto et al., 2018). We learn the reward function $r_\theta(z_t, a_t)$ using data $s_t, a_t, r_t$ from the *real environment* only. For $r_\theta(z_t, a_t)$, by minimizing the mean squared error between true and predicted rewards :

$$\mathcal{L}_{r_\theta}(z_t \sim e_\phi(\cdot \mid s_t), a_t, r_t) = (r_\theta(z_t, a_t) - r_t)^2. \tag{11}$$

Unlike prior representation learning methods (Zhang et al., 2020), we do not use the reward or Q function training signals to train the encoder. The encoder, model, and policy are optimized using the principled joint objective only.

## D   IMPLEMENTATION DETAILS

We implement ALM using DDPG (Lillicrap et al., 2015) as the base algorithm. Following prior svg methods (Amos et al., 2020), we parameterize the encoder, model, policy, reward, classifier and Q-function as 2-layer neural networks, all with 512 hidden units except the model which has 1024 hidden units. The model and the encoder output a multivariate Gaussian distribution over the latent-space with diagonal covariance. Like prior work (Hansen et al., 2022; Yarats et al., 2021), we apply layer normalization (Ba et al., 2016) to the value function and rewards. Similar to prior work (Schulman et al., 2015; Hafner et al., 2019), we reduce variance of the policy objective in Equation 8, by computing an exponentially-weighted average of the objective for rollouts of length 1 to K. This average is also a lower bound (Appendix A.3). To train the policy, reward, classifier and Q-function we use the representation sampled from the target encoder. For exploration, we use added normal noise with a linear schedule for the std (Yarats et al., 2021). All hyperparameters are listed in Table 3. The brief summary of all the neural networks, their loss functions and the inputs to their loss functions are listed in Table 4.

## E   ADDITIONAL EXPERIMENTS

**Analyzing the learned representations.**    Because the ALM objective optimizes the encoder and model to be self-consistent, we expect the ALM dynamics model to remain accurate for longer rollouts than alternative methods. We test this hypothesis using an optimal trajectory from the HalfCheetah-v2 task. Starting with the representation of the initial state, we autoregressively unroll the learned model, comparing each prediction $z_t$ to the true representation (obtained by applying the encoder to observation $s_t$). Fig. 7 (left) visualizes the first coordinate of the representation, and shows that the model learned via ALM remains accurate for $\sim 20$ steps. The ablation of ALM that removes the KL term diverges after just two steps.

**Bias and Variance of the Lower Bound.**    We recreate the experimental setup of (Chen et al., 2021) to evaluate the average and the std value of the bias between the Monte-Carlo returns and the estimated returns (lower bound $\mathcal{L}_\phi^K(s, a)$ for our experiments). Similar to (Chen et al., 2021), since

Table 3: A default set of hyper-parameters used in our experiments.

| Hyperparameters | Value |
|---|---|
| Discount ($\gamma$) | 0.99 |
| Warmup steps | 5000 |
| Soft update rate ($\tau$) | 0.005 |
| Weighted target parameter ($\lambda$) | 0.95 |
| Replay Buffer | $10^6$ for humanoid |
| | $10^5$ otherwise |
| Batch size | 512 |
| Learning rate | 1e-4 |
| Max grad norm | 100.0 |
| Latent dimension | 50 |
| Coefficient of classifier rewards | 0.1 |
| Exploration stddev. clip | 0.3 |
| Exploration stddev. schedule | linear(1.0 , 0.1, 100000) |

Table 4: Neural networks used by ALM

| Neural Network | Loss Function | Inputs to Loss |
|---|---|---|
| Encoder: $e_\phi(s) \to z$ 
 Model: $m_\phi(z, a) \to z'$ | joint lower bound (Eq. 5) | $\{s_i, a_i, s_{i+1}\}_{i=t}^{t+K-1}$ |
| Policy: $\pi_\phi(z) \to a$ | joint lower bound (Eq. 5) | $z_t$ |
| Q-Function: $Q_\theta(z, a) \in \mathbb{R}$ | TD-loss (Eq. 10) | $z_t, a_t, r_t, z_{t+1}$ |
| Classifier: $C_\theta(z, a, z') \in (0, 1)$ | cross entropy loss (Eq. 9) | $z_t, a_t$ 
 $z_{t+1} \sim e_\phi(s_{t+1})$ 
 $\hat{z_{t+1}} \sim m_\phi(z_t, a_t)$ |
| Reward Function: $r_\theta(z, a) \in \mathbb{R}$ | MSE (Eq. 11) | $z_t, a_t, r_t$ |

the true returns change as training progresses we analyze the mean and std value of normalized bias, which is defined as: $(\mathcal{L}_\phi^K(s, a) - Q^\pi(s, a))/|E_{s', a' \sim \pi}(s', a')|$. This helps us to evaluate bias relative to the scale of average Monte-Carlo returns of the current policy. Results for Ant-v2 and Walker2d-v2 are presented in Fig. 8.

**Additional Ablations.**

- In Table 5 we train ALM using the soft actor critic entropy bonus and compare it with SAC-SVG.

- In Figure 11 we show that ALM is robust to a range of coefficients for the KL term in Equation 8.

- In Figure 12, we incorporate the logarithmic transformation of the reward function based on Appendix A.8 and show that it does not hurt performance.

- In Figure 13, we compare ALM to a version of MnM Eysenbach et al. (2021a) to show that learning the encoder is necessary for good performance on high dimensional tasks.

- In Figure 14, we show that ALM achieves higher asymptotic returns when compared to SAC.

- In Figure 15, we compare ALM to a version of ALM which uses reconstruction loss to learn the encoder.

- In Figure 16, we show that ALM works well even when using a linear classifier.

- In Figure 17, we add noise in the MDP dynamics to show the performance of ALM with varying aleatoric uncertainty.

- In Figure 18 and Figure 19, we compare ALM to a version of ALM which additionally optimizes the encoder and the latent space model to predict the value function.

- In Figure 20, we show the average final episodic returns achieved by ALM for different coefficients for the KL term in Equation 8.

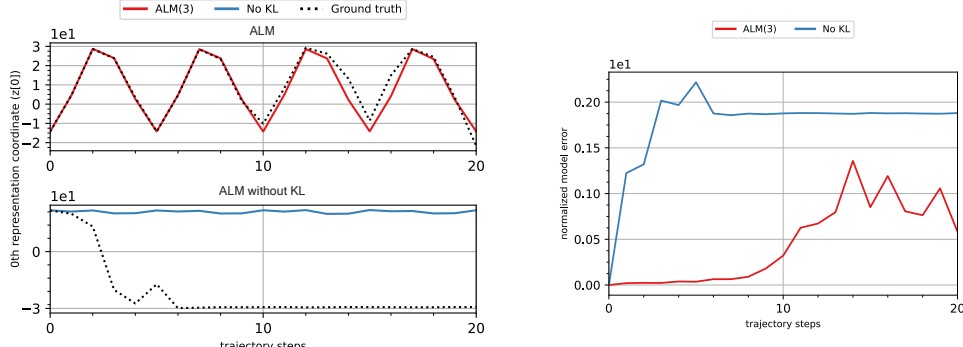

Figure 7: **Analyzing the learned representations**: *(Left-top)* The ground truth representations are obtained from the respective trained encoders on the same optimal trajectory. *(Left-bottom)* Without the KL term, the representations learnt are degenerate, i.e. they correspond to the same value for different states. *(Right)* The KL term in the ALM objective, trains the model to reduce the future $K$ step prediction errors. The latent-space model is accurately able to approximate the true representations upto $\sim 20$ rollout steps.

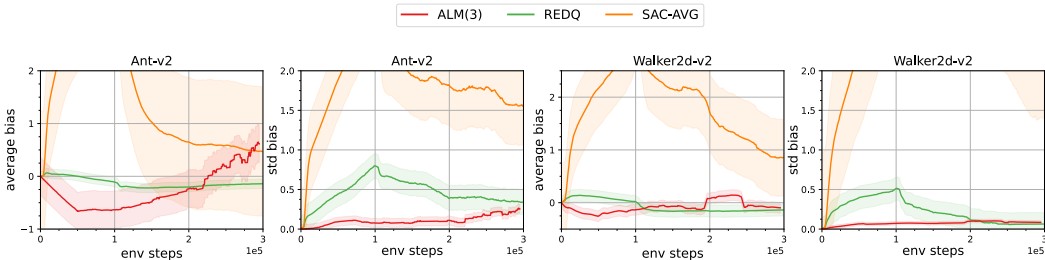

Figure 8: **Bias and Variance of the Lower Bound**: In accordance with what our theory suggests, the joint objective is a biased estimate of the true returns. The std values are uniform throughout training and consistently lower than REDQ, which could be the reason behind the sample efficiency of ALM(3)

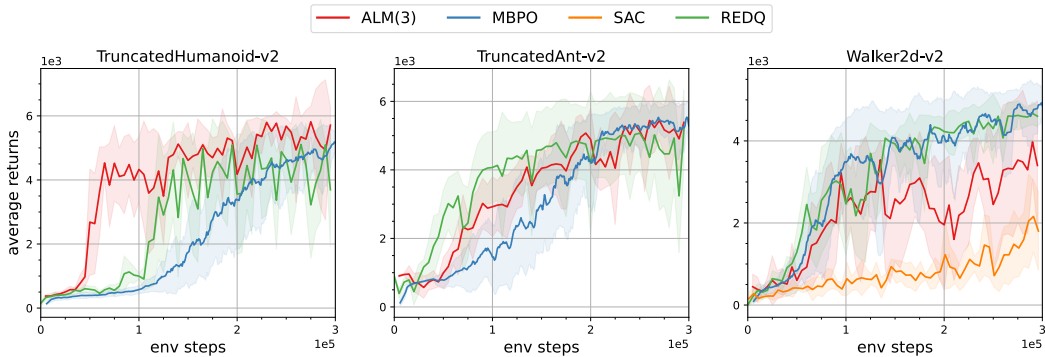

Figure 9: **Additional sample efficiency experiments.**: ALM generally matches the sample efficiency of MBPO and REDQ at a fraction of the computation complexity.

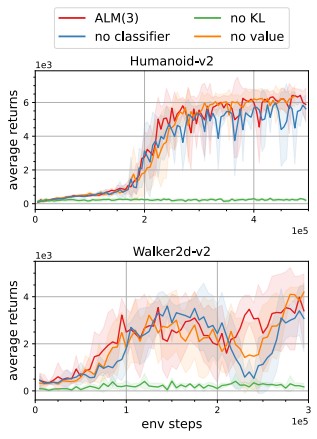
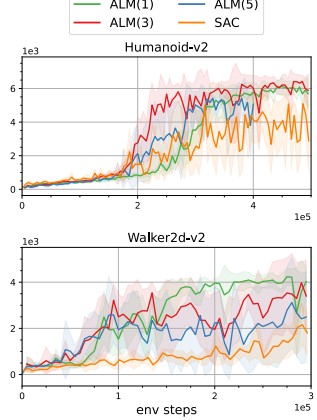
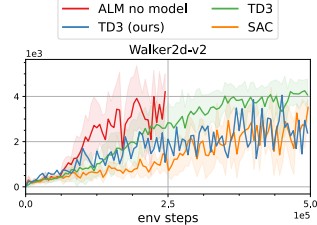

(a) Additional results for comparing different components of the joint objective on Humanoid-v2 and Walker-v2. The KL consistency term is the most important contributing factor.

(b) Additional results for comparing different horizon lengths. All horizon lengths generally perform better than sac, but horizon length 3 performs best.

(c) Additional results for comparing ALM(3)'s representations with pure model-free RL methods.

Figure 10: **Additional ablation experiments.**

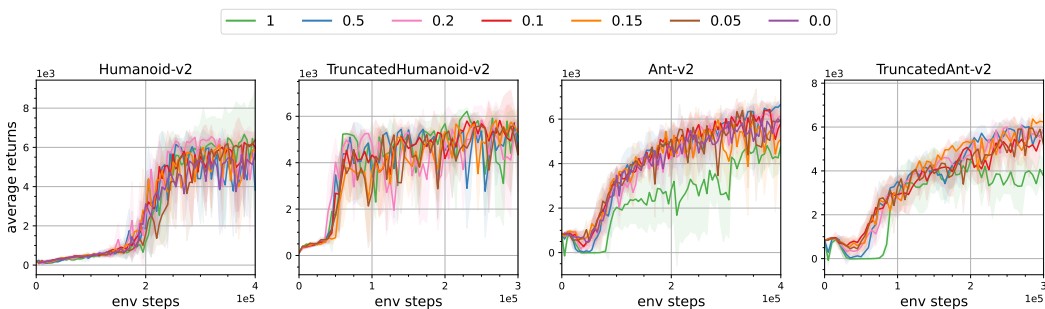

Figure 11: **ALM is robust across different values of the coefficient for the KL term in Equation 8.** In our implementation, we deviate from the value 1, because it leads to relatively gradual increase in returns on some environments. This is expected because higher coefficient to the KL term leads to higher compression Eysenbach et al. (2021b). We add that prior work on variational inference Wenzel et al. (2020) also finds that scaling the KL term can improve results.

Table 5: Where possible, we have tried to use the same hyperparameters and architectures as SAC-SVG. We use the same training hyperparameters like update to data collection ratio, batch size and rollout length when compared to sac-svg. Both methods use the same policy improvement technique: stochastic value gradients. The two exceptions are decisions that simplify our method: unlike SAC-SVG, we use a feedforward dynamics model instead of an RNN; unlike SAC-SVG, we use a simple random noise instead of a more complex entropy schedule for exploration. To test whether the soft actor critic's entropy, used in SAC-SVG can be a confounding factor causing SAC-SVG to perform worse than ALM, we compare a version of ALM which uses a soft actor critic entropy bonus like the SAC-SVG.

| Method | T-Humanoid-v2 | T-Ant-v2 |
|---|---|---|
| ALM-ENT(3) | $4747 _{\pm 900}$ | $\mathbf{4921} _{\pm \mathbf{128}}$ |
| ALM(3) | $\mathbf{5306} _{\pm \mathbf{437}}$ | $4887 _{\pm 1027}$ |
| SAC-SVG(3) | $501 _{\pm 307}$ | $5306 _{\pm 437}$ |
| SAC-SVG(2) | $472 _{\pm 85}$ | $3833 _{\pm 1418}$ |

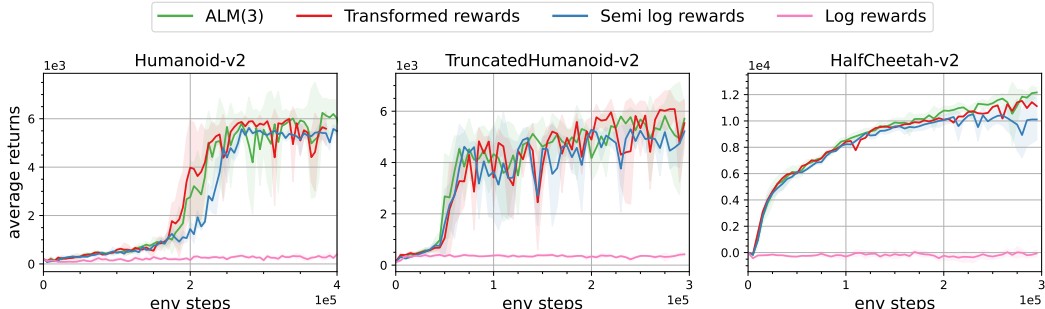

Figure 12: **Using the logarithm does not actually hurt performance.** Our theory suggests that we should take the logarithm of the reward function and Q-function. Naïvely implemented, this logarithmic transformation (pink) performs much worse than omitting the transformation (green). We also see that using a log of rewards for only training the encoder and model does not affect the performance (blue). We hypothesize that the non linearity of log(x) for reward values makes the Q-values similar for different actions. However, by transforming the reward function (which does not change the optimization problem), we are able to include the theoretically-suggested logarithm while retaining high performance (red). See Appendix A.8 for more details.

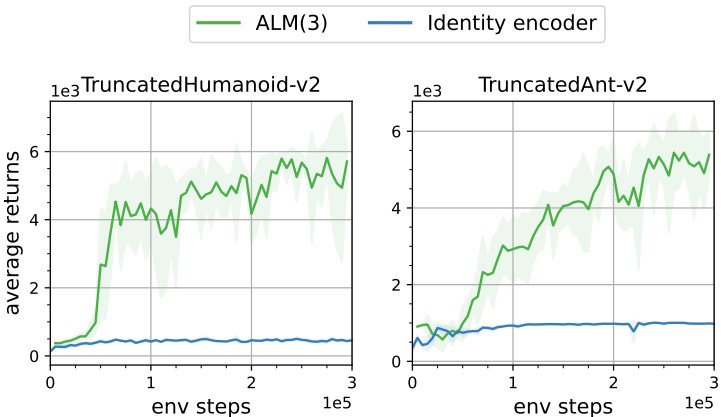

Figure 13: **Comparison with an MnM ablation.** The main difference between ALM and a prior joint optimization method (MnM), is that ALM learns the encoder function. Replacing that learned encoder with an identify function yields a method that resembles MnM, and performs much worse. This result supports our claim that RL methods that use latent-space models can significantly outperform state-space models.

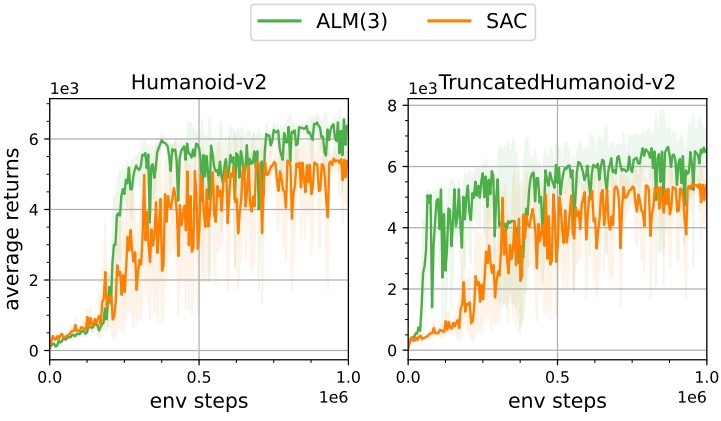

Figure 14: **Asymptotic performance.** Even after 1 million environment steps, ALM still outperforms the SAC baseline.

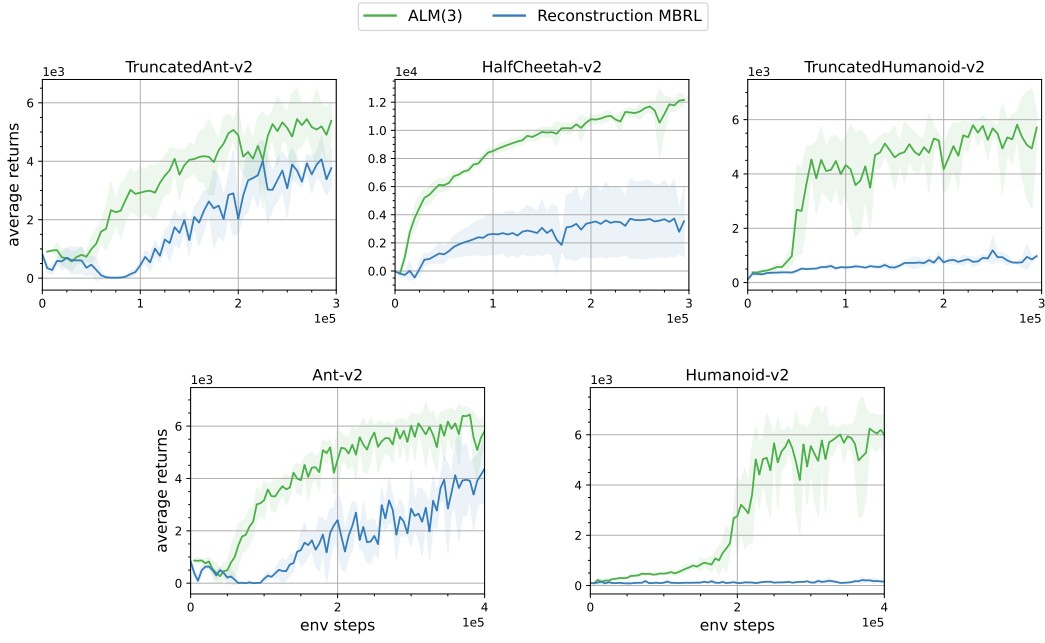

Figure 15: **Importance of an aligned objective.** Comparison with a version of ALM that learns policies using the same objective, but learns representations and latent-space models to maximize likelihood. This further verifies the claims made in Section 5.1 that using an aligned objective for joint optimization leads to superior performance

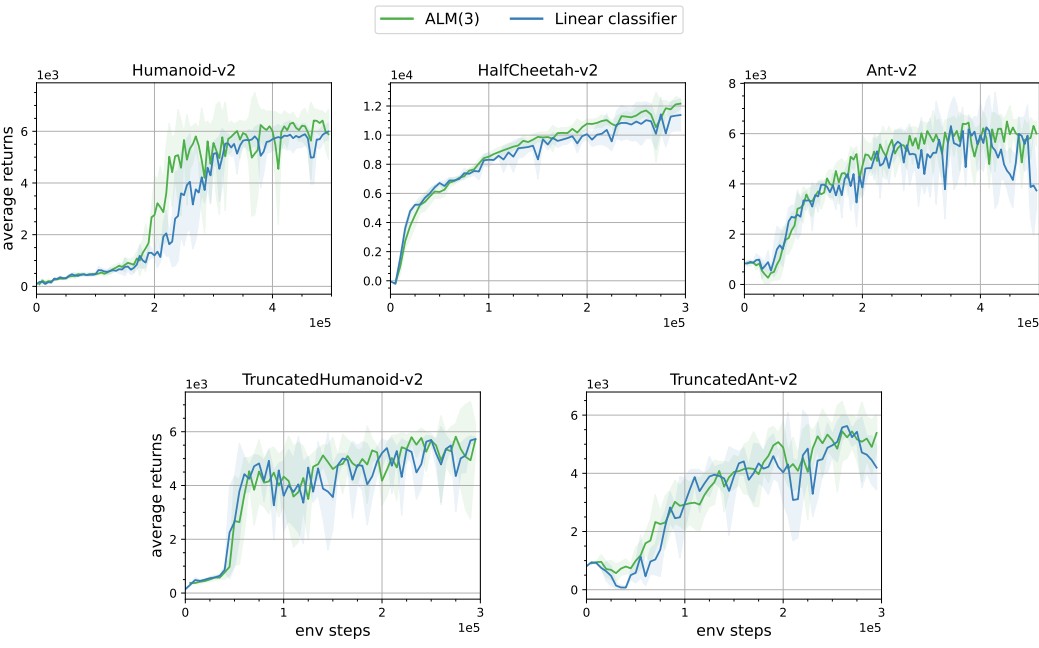

Figure 16: **Robustness to classifier errors.** ALM retains a high degree of performance even when the classifier is restricted to be a linear function, showing how ALM can be robust to errors in the classifier.

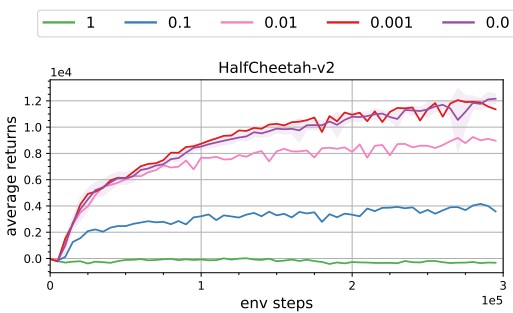

Figure 17: **Robustness towards various levels of aleatoric uncertainty.** We created a version of the HalfCheetah-v2 task with varying levels of aleatoric noise.

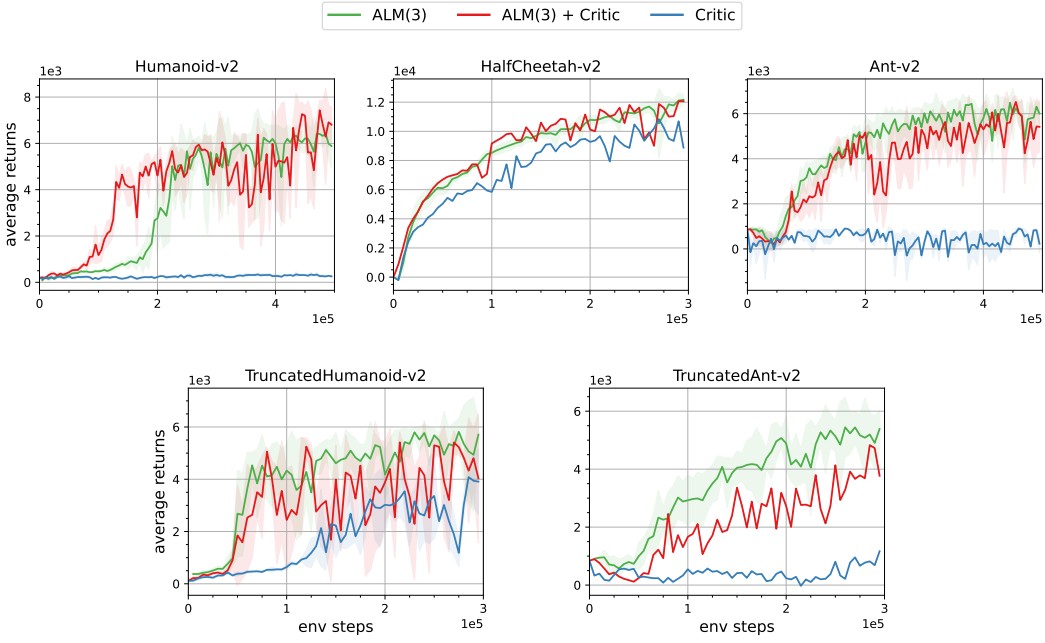

Figure 18: **Effect of learning to predict the value function.**

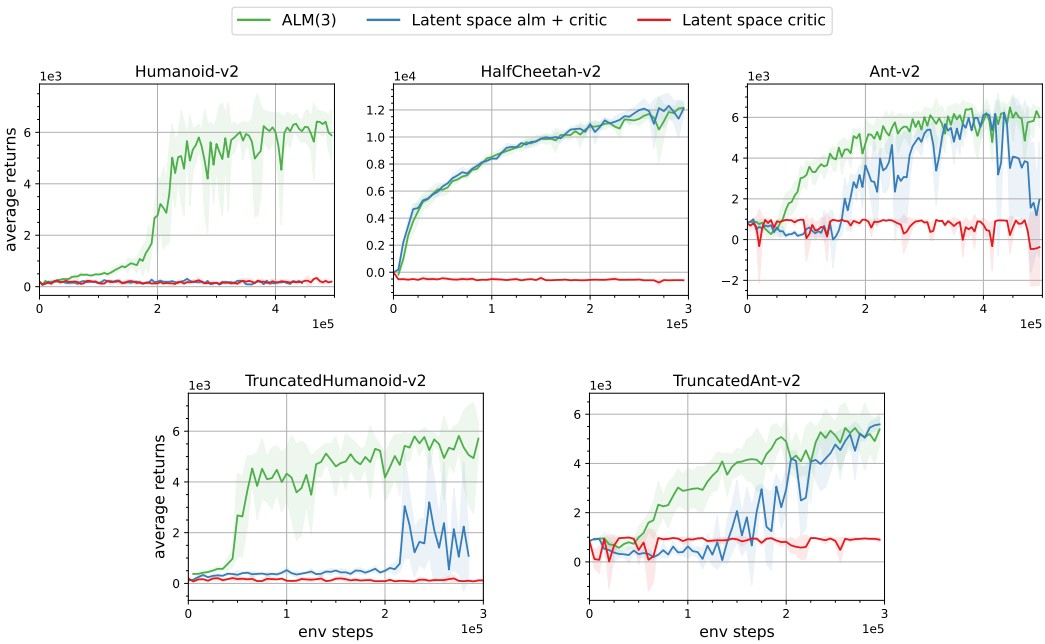

Figure 19: **Effect of learning the encoder as well as latent-space model to predict the value function.**

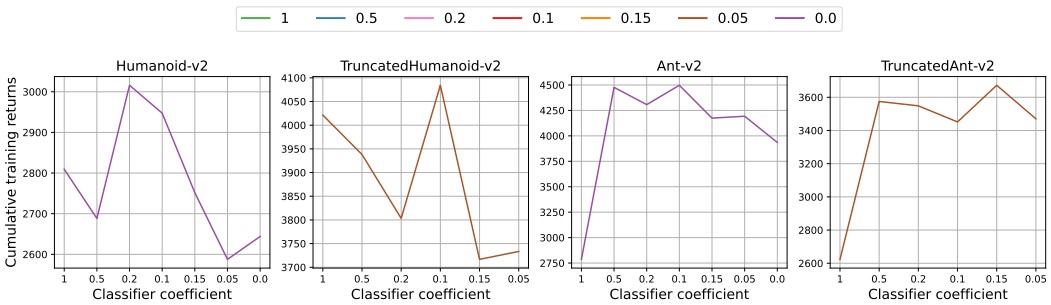

Figure 20: **Returns at different values for classifier coefficient.**

## F    COMPARISON TO PRIOR WORK ON LOWER BOUND FOR RL (MNM)

Our approach of deriving an evidence lower bound on the RL objective is similar to prior work (Eysenbach et al., 2021a). In this section we briefly go over the connection between our method and Eysenbach et al. (2021a). The lower bound presented in (Eysenbach et al., 2021a) is a special case of the lower bound in 3.1. By taking the limit $K \to \infty$ and assuming an identity function for the encoder, we exactly reach the lower bound presented in Eysenbach et al. (2021a). By using a bottleneck policy (policy with an encoder), ALM(K) learns to represent the observations according to their importance in the control problems rather than trying to reconstruct every observation feature with high fidelity. This is supported by the fact that ALM solves Humanoid-v2 and Ant-v2, which were not solvable by prior methods like MBPO and MnM. By using the model for $K$ steps, we have a parameter to explicitly control the planning / Q-learning bias. Hence, the policy faces a penalty (intrinsic reward term) only for the first K steps rather than the entire horizon (Eysenbach et al., 2021a), which could lead to lower returns on the training tasks (Eysenbach et al., 2021b).

## G    FAILED EXPERIMENTS

Experiments that we tried and that did not help substantially:

- Value expansion: Training the critic using data generated from the model rollouts. The added complexity did not add much benefit.
- Warm up steps: Training the policy using real data for a fixed time-steps at the start of training.
- Horizon scheduling: Scheduling the sequence length from $1$ to $K$ at the start of training.
- Exponential discounting: Down-scaling the learning rate of future time-steps using a temporal discount factor, to avoid exploding or vanishing gradients.

Experiments that were tried and found to help:

- Target encoder: Using a target encoder for the KL term in Eq. 7 helped reduce variance in episode returns.
- Elu activation: Switching from Relu to Elu activations for all networks for ALM resulted in more stable and sample efficient performance across all tasks.

