# OpenReview forum: "Simplifying Model-based RL: Learning Representations, Latent-space Models, and Policies with One Objective"
_ICLR.cc/2023/Conference — ICLR 2023 poster_

### Official Review · Reviewer_km6q · 2022-10-24

**Confidence:** 2
**Clarity, Quality, Novelty And Reproducibility:** Mentioned above.
**Correctness:** 3
**Technical Novelty And Significance:** 3
**Empirical Novelty And Significance:** 3
**Recommendation:** 6

**Strength And Weaknesses:**

Strengths
- Recent works (especially on offline RL) have shown that we can find simpler alternatives to current RL objectives without compromising performance. This paper adds up nicely to those works by showing that we can also simplify model-based RL objectives.
- The work proposes a principled way to train state encoders for RL.
- The proposed method has higher sample efficiency compared to the baselines.

Weaknesses
- As the authors note, it seems the method still requires a few tweaks to work well empirically. For example, we need to omit the log of the true rewards and scale the KL term in the policy objective to 0.1. While the authors provide a brief intuition on why those modifications are needed, I think the authors should provide more concrete analysis (e.g., empirical results) on what problems the original formula have and how the modifications fix them. Also, it would be better if the authors provide ablation results on those modifications. For example, does the performance drop drastically if the scale of the KL term changes (to 0.05, 0.2, 0.5, ...) ?
- The compute comparison vs. REDQ  on Figure 3 seems to be misleading. First, less runtime does not necessarily mean less computational cost. Second, if the authors had used the official implementation of REDQ [1], it should be emphasized that this implementation is very inefficient in terms of runtime. In detail, the implementation feeds forward to each Q-network one at a time while this feed-forward process is embarrassingly parallelizable. The runtime of REDQ will drop significantly if we parallelize this process.
- The ablation experiments seem to show that the value term for the encoder is not necessary. It would be better to provide an explanation on this result.
- Some of the equations seem to have typos. 1) On equation (4), the first product and the second product have exactly the same form. 2) On algorithm 1 Line 8, shouldn't we use s_n instead of s_t?

Questions
- I am curious of the asymptotic performance of the proposed method. If possible, can the authors provide average return results with more env steps?

[1] https://github.com/watchernyu/REDQ

**Summary Of The Paper:**

The paper proposes a new objective for model-based RL that lower bounds the true objective and can be used to optimize the encoder, model, and the policy at once. The proposed model shows higher sample efficiency compared to the baselines.

**Summary Of The Review:**

The paper proposes a nice approach to unify (and simplify) model-based RL objective. With some additional few tweaks, the proposed methods shows high empirical performance compared to the baselines.

---

> ### Author Response · Authors · 2022-11-13
> **Response to km6q**
>
> Dear reviewer,
>
> **[response 1 of 2]** We thank the reviewer for their detailed feedback. It seems like the reviewer's main concerns are the absence of concrete empirical results to support the changes from theory, compute comparison vs. REDQ and asymptotic returns.  We add 4 new ablation experiments to address these concerns:
>
> 1) In figure 11 (Pg 26), we show that ALM performs similarly, at a range of values of the KL coefficient (0.0, 0.05, 0.15, 0.1, 0.2, 0.5, 1) across 4 tasks. This shows that ALM is not overly sensitive to  KL coefficients.
> 2) In figure 12 (Pg 26), we show that adding the log of rewards / Q function hurts the performance of ALM, thus motivating its omission. We also added a new version of ALM which is able to retain the empirical performance, while using the log of rewards / Q function, thus solving the issue of deviation from the log scale.
> 3) In figure 14 (Pg 27), we outperform SAC, even after 1 million timesteps on both tasks considered.
> To address reviewer’s concern about the parallelizability, we optimized the ensemble training in the official REDQ code to be parallelized, leading to an increase in training speeds by 3 times, which is still 2 times slower than our method.
>
> Below, we will describe these experiments in more detail and answer the specific reviewer questions **Do these new experiments, revisions and answers address all the reviewer's concerns?**
>
> > it would be better if the authors provide ablation results on those modifications
>
> The practical version of our algorithm makes two changes:
>
> 1) Using a different KL coefficient.
> In our new Figure 11 (Pg 26), we perform an extensive ablation for 5 different values (7 if you include no classifier ablation, coeff=0, and the original ALM, coeff=0.1) across 4 tasks. From Figure 11, we find that ALM is robust to different values of KL coefficient. We choose KL coefficient of 0.1 instead of 1 because it leads to relatively slower training in the Ant environments. We add that prior work on variational inference [1,2] also finds that scaling the KL term can improve results.
>
> 2) Omitting the log of rewards.
> To provide new theoretical evidence, In appendix D1,  we show that omitting the log of rewards/Q-function, for all practical purposes, is equivalent to scaling the KL term. We show that omitting the log is like applying log to a transformed MDP which has the same optimal policy as the original MDP. Based on this we perform an ablation showing that the log of rewards / Q-function can be added back without changing the performance.
>
> To provide empirical evidence, In our new Figure 12  (Pg 26), we run three versions of ALM across 2 tasks:
> 1) Semi log reward: We add a log for rewards and Q function only for the representation and model objective (equation 7).
> 2) Log rewards: We add a log for all three representation, model and policy objectives. (equation 7 and 8).
> 3) Transformed log reward: Transforms the original MDP by adding a large constant (=10000) to all rewards. Note that this does not change the RL optimization problem. We then add a log for rewards and Q function in this transformed MDP for all three  representation, model and policy objectives. (equation 7 and 8).
>
> The results, shown in Figure 12, show that  Transforming the MDP and then adding the log of rewards as suggested by the theory, retains the high empirical performance. Without the transformed MDP, we also see that using a log of rewards for training the encoder and model does not affect the performance, but adding a log of rewards for training the policy does not work. In the non transformed MDP, we hypothesize that the non linearity of log(x) for lower reward values makes the Q-values similar for different actions. This in turn makes it difficult for the actor’s loss function flat and causes the gradients to vanish.
>
> > The runtime of REDQ will drop significantly if we parallelize this process.
>
> As suggested by the reviewer, we implemented a version of REDQ that does update the ensemble in parallel, using PyTorch's vmap function. This implementation was 3 times faster than REDQ, which is still 2 times slower than our method (which does not employ parallelization optimization).
>
> > I am curious of the asymptotic performance of the proposed method. If possible, can the authors provide average return results with more env steps?
>
> We trained ALM of Humanoid-v2 and TruncatedHumanoid-v2 for 1 million time steps. We have added this result in our new figure 14 (Pg 27). ALM achieves higher sample efficiency as well as higher asymptotic returns on both the tasks when compared to soft actor critic.
>
> [1] Wenzel, Florian, et al. "How Good is the Bayes Posterior in Deep Neural Networks Really?"
>
> [2] Higgins, Irina, et al. "beta-VAE: Learning Basic Visual Concepts with a Constrained Variational Framework"

---

> > ### Author Response · Authors · 2022-11-13
> > **Response to km6q**
> >
> > **[response 2 of 2]**
> > > First, less runtime does not necessarily mean less computational cost
> >
> > We have revised figure 3 (Pg 7) and its caption to refer to the ``runtime'' of these methods, rather than the compute cost. In the bar plot in figure 3 (bottom right), we compare and show that the REDQ is 6 times slower to complete an environment step when compared with ALM.
> >
> > > The ablation experiments seem to show that the value term for the encoder is not necessary. It would be better to provide an explanation on this result.
> >
> > We have added a line to the last paragraph of section 5.3 to explain that good results can still be achieved if this part of our theoretically-derived method is removed. Note, however, that our theory also motivates the KL term, which is very important for achieving good results (see Fig. 5 a). We hypothesize that the value term may not be necessary because its effect, driving exploration, may already be incorporated by Q-value overestimation (a common problem for RL algorithms [RL textbook, td3]).
> >
> > > Some of the equations seem to have typos.
> >
> > We have fixed these typos. For the 8th line in algorithm 1, it should be s_t. Because the state to calculate policy objective (equation 9) is sampled from the buffer, we have changed the notation to make it clearer.

---

> > ### Comment · Reviewer_km6q · 2022-11-16
> > **Response to Authors**
> >
> > Thanks for the detailed response.
> >
> > I have read the responses including those to other reviewers. The additional experiments address most of my concerns. While I do think that deriving a unified objective for model-based RL is itself a valuable contribution, I also agree with reviewer RXsd that deviation of theory and practice can make the derived theory less meaningful. I appreciate the authors' effort to incorporate the original log operations (section D.1.), however, the suggested reward shifting (up to + 10000) seems somewhat ad-hoc and also this would not apply to environments with terminations.
> >
> > Considering these, I am raising my score to marginal accept, but with low confidence.

---

> > > ### Author Response · Authors · 2022-11-16
> > > **Response to km6q**
> > >
> > > Dear reviewer,
> > >
> > > Thanks for confirming that the response and experiments addressed the concerns. We are happy to answer any more questions that you might have.

---

### Official Review · Reviewer_PJmH · 2022-10-25

**Confidence:** 4
**Correctness:** 4
**Technical Novelty And Significance:** 3
**Empirical Novelty And Significance:** 3
**Recommendation:** 6

**Clarity, Quality, Novelty And Reproducibility:**

To the best of my knowledge, the proposed objective is novel. The paper and the code are well-written and organized. I believe the results are reproducible thanks to the paper and the code.

**Strength And Weaknesses:**

In model-based RL algorithms like SAC-SVG (Amos et al., 2020 [2]) one needs to train an actor as well as a critic in a way that actor-model output should be the input to the critic-model to enable backpropagation to the actor through the critic. This makes the boundaries between an actor and a critic vague. For example, the representation model and dynamic model can be parts of the actor and optimized with actor loss, or be parts of the critic and optimized with critic loss. In more complex cases these models can be jointly optimized with critic and actor losses, or even with some auxiliary losses (such as state reconstructions), or with some combinations of the before-mentioned losses.

Orthogonal to these choices is the decision whether to use evidence lower bound on these objectives or not. As far as I know, this paper is the first work that successfully explores the possibility of training a representation model and a dynamic model only with actor loss and it did so using evidence lower bound.

The authors discuss that the proposed objective leads to self-consistency for the actor. However, I want to point out that the paper's objective leads to inconsistency for a critic as the authors train the critic and the reward subnetwork on top of representations that are optimized for the actor and not for the critic. Plus, I believe it is an open question whether the actor's consistency or the critic's consistency is more important, or whether one needs to jointly optimize representation with the actor's and the critic's objectives. It would be great if the authors can comment on this. Plus, I recommend adding experiments if it is possible:
1) try to optimize the representation model and the dynamic model with critic and reward losses but optimize the policy model with objective (8) derived in the paper.
2) try to optimize the representation model and the dynamic model with the actor (objective (8)), critic, and reward objective jointly.
3) the same as (1) and (2) but change only the representation model objective as it seems to be easier with the current coding.

In the paper, the main baseline (SAC-SVG) uses an auxiliary state-reconstruction objective and performs worse than the proposed method. However, in my opinion, the actor objective is much more aligned with the critic objective than with auxiliary objectives such as state-reconstruction objectives or SSL-objective. So I miss a comparison with the strong baseline that uses only actor and/or critic objectives to train model parts (e.g. with [1]).

Strength
1) novel evidence low bound on RL objective is derived;
2) novel loss is well-explained, motivated, discussed, and analyzed. (E.g. that the loss encouraged mode-seeking behavior)
3) The method shows that axillary objectives (such as state reconstruction) may be hurtful to the performance.
4) The paper is well-written;
5) The authors show that the proposed algorithm beats similar SAC-like algorithms that use auxiliary objectives as state reconstruction to learn representations. Plus, the proposed method is less computationally demanding than methods that use ensembles (MBPO, REDQ), but archives similar or better performance.
6) An ablation study is present.

Minor issues and questions
1) Authors claim that the proposed method is robust to the number of rollouts, but in Fig.5b (bottom) ALM(5) performance is quite low.
2) Why not use the same parametrization, objective (8) with the classifier, to train the encoder and dynamics model?
3) Maybe a small table that shows the main differences between the proposed algorithm and the baselines would be helpful;
4) In the introduction, the authors mention that MuZero uses value prediction to train a representation model and a dynamic model. I would like to point out that MuZero uses not only value prediction loss but also use the loss for the policy model (actor objective) to train these models. So representation model and a dynamic model are jointly optimized with actor and critic losses in MuZero and I argue this makes representation consistent with both actor and critic objectives.

[1] Hubert et al, Learning and Planning in Complex Action Spaces, 2021.

[2] Amos et al, On the model-based stochastic value gradient for continuous reinforcement learning, 2020.


**Summary Of The Paper:**

The authors propose to train a policy, a state-encoder, and a dynamic model in model-based RL with the novel evidence lower bound on expected returns loss. In this way policy, encoder, and dynamic model have the same objective to maximize which differs this work from the previous ones.


**Summary Of The Review:**

Overall the paper is well-written and motivated. The experimental results are very interesting. However, in my opinion, some key experiments and baselines (e.g. [1] Learning and Planning in Complex Action Spaces) are missing.

---

> ### Author Response · Authors · 2022-11-13
> **Response to PJmH**
>
> Dear Reviewer,
>
> We thank the reviewer for the detailed review, and for the feedback on the paper. Our understanding is that the main suggestion is about _also_ jointly optimizing the critic with the same objective as the other components, which we have investigated using variants of ALM with two new representation learning objectives:
>
> [V-1] Representations are learnt to minimize the critic and the reward loss (equations 10, 11).
> [V-2] Representations are learning using ALM objectives as well as the critic and reward loss.
>
> In figure 18 (Pg 29), variant [V-1] performs worse on all 5 tasks. Variant [V-2] leads to unstable returns on 3 tasks and increases sample efficiency on 1. Note that for both variants, the model and the policy are trained using ALM.
>
> **Have these new experiments, as well as the revision to the paper, fully addressed the concerns about the paper?**
>
> > Incorporating the critic loss into the objective
>
> As suggested by the reviewer, we explored two ways of incorporating the critic loss into the ALM objective, variants [V-1] and [V-2] described above. These ways are inspired by the reviewer's suggestions #3.
>
> The results, shown in figure 18 (Pg 29), show that just using the critic loss ([V-1]) to learn representations is worse than ALM on all tasks. Adding the critic loss to ALM representation learning improves sample efficiency in 1 task, while increasing variance in 4.
>
> > actor's consistency or the critic's consistency is more important; It would be great if the authors can comment on this
>
> The new experiments studying critic consistency (figure 18) show that incorporating critic consistency can decrease performance, suggesting that actor consistency is more important than critic consistency.
>
>
> > So I miss a comparison with the strong baseline that uses only actor and/or critic objectives to train model parts (e.g. with [1]).
>
> The code for (Hubert et al) is not open sourced. We would be happy to compare additional open-source baselines. We note that our current paper already compares with 7 baselines, which is more than prior work (e.g., MBPO compares with 5 baselines, MPPI compares with 1 baseline).
>
> **Do these experiments address the reviewer's questions about alignment with the critic objective?** If not, we would be happy to run more experiments.
>
>
>
> > Authors claim that the proposed method is robust to the number of rollouts, but in Fig.5b (bottom) ALM(5) performance is quite low.
>
> We have removed this claim about robustness from Section 5.3.
>
> > Why not use the same parametrization, objective (8) with the classifier, to train the encoder and dynamics model?
>
> Training the encoder and dynamics model with Eq. 8 would require an on-policy algorithm, which likely would decrease the sample efficiency of our method.  For off-policy actions sampled from the replay buffer, we have access to the next state and therefore can calculate the KL divergence directly without using a classifier based approximation. Additionally, generative models learnt using classifiers are more difficult to train. An on-policy version of ALM, where objective (8) can be calculated and optimized with the encoder, model and policy, together, without using a classifier is possible. This is certainly an interesting direction and we have added it as a direction for future work in Section 6, under limitations and future work.
>
> > Maybe a small table that shows the main differences between the proposed algorithm and the baselines would be helpful;
>
> We have added this table to Appendix F (Pg 30).
>
> > In the introduction, the authors mention that MuZero uses value prediction to train a representation model and a dynamic model. I would like to point out that MuZero uses not only value prediction loss but also use the loss for the policy model (actor objective) to train these models. So representation model and a dynamic model are jointly optimized with actor and critic losses in MuZero and I argue this makes representation consistent with both actor and critic objectives.
>
> We have revised this sentence to say that the representations for MuZero are trained using both the actor and the critic objectives. This is different from our method, which also trains these representations to be more easily predictable.

---

> > ### Comment · Reviewer_PJmH · 2022-11-14
> > **Response to the Authors**
> >
> > Dear Authors,
> >
> > thank you for your responses and the great additional ablation study!
> >
> > I have a few additional notes:
> > 1) In the code you do not use discounting factor to train the model and the encoder. The code differs from objective (7) in the paper in this respect.
> > 2) It would be great to see ablation results where the model and the encoder are trained only with KL loss (just by removing reward and Q-value terms from objective (7)). In this way, the model and encoder objective becomes self-supervised and resembles BYOL[1]. I believe it is an easy way to ablate your model against a SSL objective and estimate the importance of the KL term.
> >
> > [1] Grill et al., Bootstrap your own latent: A new approach to self-supervised Learning, 2020

---

> > > ### Author Response · Authors · 2022-11-14
> > > **Response to PJmH**
> > >
> > > Response to PJmH
> > >
> > > 1) We have added a footnote on Page 5 to state that we have omitted the gamma discounting from our code. We have run the experiments with gamma discounting right now.
> > >
> > > 2) In figure 5, we have run this exact ablation (See Figure 10 for this ablation on more tasks). Our results show that the temporal consistency term is important for good performance. We have run this ablation for additional tasks.
> > >
> > > We have also coded an ablation where both the encoder and the model are trained using 1) ALM loss + critic loss 2) Only critic loss. Because of limited compute, we are not sure whether our ablations will be completed in time.
> > >
> > > Together with the new experiment, which are running now, we believe that the revised paper is substantially stronger than the original submission, thanks to the additional experiments and feedback from the reviewers. We believe that the revised version has addressed all the concerns mentioned by the reviewer, and welcome the reviewer to re-evaluate the paper in light of these revisions.

---

> > > > ### Comment · Reviewer_PJmH · 2022-11-15
> > > > **Response to Authors**
> > > >
> > > > > In figure 5, we have run this exact ablation (See Figure 10 for this ablation on more tasks). Our results show that the temporal consistency term is important for good performance. We have run this ablation for additional tasks.
> > > >
> > > > Thank you for pointing this out. I did not realize before that `no value` means that you train the world model and the encoder only with SSL-objective.
> > > >
> > > > My main concern now is that to archive the same high performance as ALM it is enough to train two components (encoder and world model) with SSL BYOL-like loss and train policy without KL term. This can be concluded based on your ablation study (Fig. 5, Fig 10, and Fig 11). *So the model without alignment with the actor performs on par with the model with this alignment.* One can conclude the method works so well only because it leverages this BYOL-like objective.
> > > > I believe if you can not show the opposite (that this alignment with the actor objective indeed helps), you should highlight this limitation in the text so the reader can easily pick up this information.
> > > >
> > > > > The code for (Hubert et al) is not open sourced. We would be happy to compare additional open-source baselines. We note that our current paper already compares with 7 baselines, which is more than prior work (e.g., MBPO compares with 5 baselines, MPPI compares with 1 baseline).
> > > >
> > > > There is open-sourced code for EfficientZero (https://github.com/YeWR/EfficientZero). The paper works with a continuous setting by discretizing an action space.
> > > >
> > > > Plus, I agree with the review *RXsd* that KL term should be reversed. Now it is colored in red but not reversed in the text.

---

> > > > > ### Author Response · Authors · 2022-11-15
> > > > > **Response to PJmH**
> > > > >
> > > > > Dear Reviewer,
> > > > >
> > > > > Thank you for clarifying the remaining concern, which we have addressed by revising the paper (details below). **Does this address the reviewer's remaining concerns?**
> > > > >
> > > > > > My main concern now is that to archive the same high performance as ALM it is enough to train two components (encoder and world model) with SSL BYOL-like loss and train policy without KL term.
> > > > >
> > > > > We have revised the paper to highlight this limitation (page 8, last paragraph, red text).
> > > > >
> > > > > As suggested by the reviewer initially, In our new figure 19 (Pg 30), we have compared ALM to 2 variants:
> > > > >
> > > > > [V-3] Representations and latent-space models are learnt to minimize the critic and the reward loss.
> > > > > [V-4] Representations and latent-space models learning using ALM objectives as well as the critic and reward loss.
> > > > >
> > > > > In figure 18 (Pg 29), variant [V-1] performs worse on all 5 tasks. Variant [V-2] leads to unstable returns on 3 tasks. Note that for both variants, the model and the policy are trained using ALM. We believe that Figure 18 and 19 address the reviewer’s initial concerns about comparison with a strong baseline that uses only actor and/or critic objectives to train model parts.
> > > > >
> > > > >
> > > > > > There is open-sourced code for EfficientZero (https://github.com/YeWR/EfficientZero). The paper works with a continuous setting by discretizing an action space.
> > > > >
> > > > > We will attempt to add this additional baseline for the camera-ready version. As it is not set up to run on benchmark Mujoco tasks we used in our experiments, it will take a bit of time to make sure that it is working so that we can give the baseline the strongest possible footing.
> > > > >
> > > > > > So the model without alignment with the actor performs on par with the model with this alignment.
> > > > >
> > > > > This is not quite correct: while making the model aligned with the policy (by adding the value function) does not improve performance, making the policy aligned with the model (by adding the classifier term, so it avoids states where the model is inaccurate) is important for getting good performance (see Fig 5a, Ant environment).
> > > > >
> > > > >
> > > > > > Plus, I agree with the review RXsd that KL term should be reversed. Now it is colored in red but not reversed in the text.
> > > > >
> > > > > We have reversed the direction of this KL in the text.

---

> > > > > > ### Comment · Reviewer_PJmH · 2022-11-15
> > > > > > **Response to Authors**
> > > > > >
> > > > > > > This is not quite correct: while making the model aligned with the policy (by adding the value function) does not improve performance, making the policy aligned with the model (by adding the classifier term, so it avoids states where the model is inaccurate) is important for getting good performance (see Fig 5a, Ant environment).
> > > > > >
> > > > > > This happens only on Ant-v2 task (taking into account Fig.11 as well as Fig.5). Can you reproduce this effect on other tasks (maybe with longer training )?

---

> > > > > > > ### Author Response · Authors · 2022-11-15
> > > > > > > **Response to PJmH**
> > > > > > >
> > > > > > > > This happens only on Ant-v2 task
> > > > > > >
> > > > > > > Fig 5a (top) shows that the same effect also holds for the HalfCheetah-v2 environment. Fig 10a (top, bottom) show that the same holds for the Humanoid-v2 and Walker2d-v2 environments. We have also added a new sensitivity plot in Figure 20 (Pg 30). X axis is the classifier coefficient and Y-axis is the cumulative returns of the entire training run(area under the plot).  While we find that including it with a very large coefficient of 1 results in too much regularization (especially on the Ant task), this is in line work prior work on variational inference ([1,2]). We find that a coefficient > 0 does better than 0 for all four tasks. The difference is especially pronounced on the Humanoid-v2 and Ant-v2 tasks (even TruncatedHumanoid-v2), perhaps explaining why prior model-based methods have failed to solve this task [1]"
> > > > > > >
> > > > > > > Kindly let us know if this addresses the reviewer's concerns.
> > > > > > >
> > > > > > > [1] Wenzel, Florian, et al. "How Good is the Bayes Posterior in Deep Neural Networks Really?"
> > > > > > >
> > > > > > > [2] Higgins, Irina, et al. "beta-VAE: Learning Basic Visual Concepts with a Constrained Variational Framework"
> > > > > > >
> > > > > > > [3] Janner, Michael, et al. " When to Trust Your Model: Model-Based Policy Optimization"

---

> > > > > > > > ### Comment · Reviewer_PJmH · 2022-11-17
> > > > > > > > **Response to Authors**
> > > > > > > >
> > > > > > > > Dear Authors,
> > > > > > > >
> > > > > > > > thank you for the clarification and additional experiments that align model parts with a critic loss. I agree now that classifier loss indeed helps.
> > > > > > > >
> > > > > > > > >>My main concern now is that to archive the same high performance as ALM it is enough to train two components (encoder and world model) with SSL BYOL-like loss...
> > > > > > > >
> > > > > > > > > We have revised the paper to highlight this limitation (page 8, last paragraph, red text).
> > > > > > > >
> > > > > > > > Despite the successful attempt to optimize the encoder, the world model, and policy jointly with one objective, practically in your experiments, it is enough to optimize the encoder and the world model with a pure SSL objective and the policy with an objective (8) to achieve the same performance. The second variant is simpler in the code and uses fewer computes. -- I suggest adding information about this to the limitation section.

---

> > > > > > > > > ### Author Response · Authors · 2022-11-17
> > > > > > > > > **Response to PJmH**
> > > > > > > > >
> > > > > > > > > Dear reviewer,
> > > > > > > > >
> > > > > > > > > As suggested, we have revised the paper to make note of this limitation in the paper (2nd paragraph of the conclusion; red text).
> > > > > > > > >
> > > > > > > > > **Does the reviewer have any additional concerns?**

---

### Official Review · Reviewer_11kg · 2022-10-26

**Confidence:** 5
**Correctness:** 4
**Technical Novelty And Significance:** 3
**Empirical Novelty And Significance:** 3
**Recommendation:** 8

**Clarity, Quality, Novelty And Reproducibility:**

Its a very clear and well derive paper.

The quality of the paper also its on the top on the domain with clear mathematical derivations and explanations.

The code is reproducible because there is a public website already with the code available. <<REMOVED BY PCs>> (which actually bring the problem of blind review)

The losses are novel, the idea of one objective is novel.





**Strength And Weaknesses:**

Very strong derivation on the objective and losses. End-to-end learning also is another strength of the paper. High sample efficacy for high dimensional states also is a strength. Outperforming model free RL on high dimensional states also seems to be novel, without using pre-training or other self-supervised technique,

Weakness: Classifier training time seems not to be taken in account, also if there are situation where learning the classifier is difficult and how this could impact the overall algorithm. Automatic K selection would be a good addition into the algorithm.

It would interesting to see the performance of algorithm with different domains that capture different levels of aleatoric uncertainty.



**Summary Of The Paper:**

A theoretical sound model based rl algorithm derivation that can process high dimensional inputs (i.e. images) that derive one objetive function to learn the latent space of the observations, the model and the policy, which usually is done using separate objectives. This work also provides a practical algorithm, that was shown empirically to have. greater sample efficiency that SOTA. The work also provide a lower bound on the RL loss.

**Summary Of The Review:**

This work derive a lower bound on the RL loss with a strong theoretical , well supported, derivation, constructing a practical model based RL algorithm, that have higher sample efficiency than SOTA, its end-to-end learning solution, that can handle high dimensional state spaces learning the a latent representation for this states,

---

> ### Comment · Program_Chairs · 2022-11-07
> **comment by PCs**
>
> PCs removed a link revealing author identity in the above review. Note that it's violation of ICLR policy to reveal authors' identity. We believe we've already edited this review once -  we would be removing this review entirely if reviewers post the link of author identity again.

---

> ### Author Response · Authors · 2022-11-13
> **Response to 11kg**
>
> Dear reviewer,
> We thank the reviewer for the thoughtful review, as well as for the suggestions for improving the paper. We ran two new experiments to answer the reviewer's questions. Please let us know if there are any additional concerns or questions.
>
> > Classifier training time seems not to be taken in account
>
> Our comparisons of the training times in figure 3 (bottom row; Pg 7) do take into account the classifier training time. We have revised the caption to mention this.
>
> > if there are situation where learning the classifier is difficult and how this could impact the overall algorithm
>
> To study how errors in the classifier impact the overall algorithm, we ran an ablation of our method where the classifier is restricted to be linear, so that it is likely to incur errors. We compared this ablation to our full method in new figure 16 (Pg 28). The results show that on 3 out of 5 mujoco environments, the linear classifier version performs at par with ALM(3), and on the 2 Ant tasks it leads to relatively higher variance in the final returns.
>
> > Automatic K selection would be a good addition into the algorithm.
>
> We will investigate this. In particular, we will see whether the mechanism from [1] can be used to automatically determine the length of the rollout.
>
> > performance of algorithm with different domains that capture different levels of aleatoric uncertainty.
>
> To study this question, we created a version of the HalfCheetah-v2 task with varying levels of aleatoric noise – we added Gaussian noise to the state observations. We measured the performance of ALM as we increased the amount of noise. The results, shown in new figure 17 (Pg 29), show that ALM performs well at noise scale less than 0.01. At higher levels of noise, the performance starts to deteriorate.
>
> [1] Buckman, Jacob, et al. "Sample-efficient reinforcement learning with stochastic ensemble value expansion." Advances in neural information processing systems 31 (2018).

---

### Official Review · Reviewer_JaMR · 2022-10-26

**Confidence:** 3
**Correctness:** 3
**Technical Novelty And Significance:** 3
**Empirical Novelty And Significance:** 3
**Recommendation:** 6

**Clarity, Quality, Novelty And Reproducibility:**

### Clarity

This paper is written in clear language. Some points:
- In the last paragraph of Page 2, second sentence,  could you write how a sentence about how the methods mentioned (decision-aware loss functions, etc) tackle the objective mismatch problem? This is so that it is clear how they don't help with latent-space models.
- The paper introduces the basics of RL in Sectino 3.1. However, in Section 3.2 and 3.3, it mentions Q-functions and Q-values without mentioning / defining it before hand.
- Similarly, the notation Q in equatino 5 is introduced without any definition.
- I think details about the classifier used in Section 4 should be moved to the main text.
- I am confused about the  the second line of Equation 4. How is this infinite product carried out in practice? From Algorithm 1, it seems only the first K terms are used. If so, why is this part in Equation 5?
- In MBRL, I thought that the dynamics model is used to sample data that can be used for training the policy. Does this method do this? I might be missing something, but from Algorithm 1, I can only see that samples are taken from the environment? Is this right?

### Quality

This paper is generally of good quality. Some questions:
- Does the Soft-actor critic nature of the main baseline affect any of the conclusions? Since Section 5.1 mainly compares with SAC-SVG, are there any other differences between the methods that could lead to the differences in performance?
- For Section 5.1, could it be possible to compare against itself, where the model-based components are trained separately?

### Novelty
To the best of my knowledge, this is a novel paper. However, I haven't carried out an in-dept literature review recently.

### Reproducibility
Code is provided as an open source implementation. I haven't tried this code out, but being available helps with reproducibility.

**Strength And Weaknesses:**

### Strengths
- The theory is built nicely, by introducing the notation first, followed by a high level outline before diving into the maths.
- The technique appears to have some good benefits.
- Discussions are nice, with good explanations of points made.
- Good literature review.
- It was good that network hyperparameters were chosen to be consistent with baselines.

### Weaknesses
- See questions below.

**Summary Of The Paper:**

This paper builds on the latent variable view of reinforcement learning (RL) to provide a model-based RL method that jointly learns representations of observations, their transition dynamics (the representations), and the optimal policy.

It introduces the Aligned Latent Model objective, which is derived from the typical latent-variable view of RL, and a particularly defined variational distribution. This is then used in a DDPG style RL algorithm.

Experiments compare against several baselines, including SAC-SVG on several complex tasks. Key questions regarding the usefulness of the contribution is analysed using appropriate experiments, including ablations.

**Summary Of The Review:**

In general, the paper is written well, and introduces its concepts cleanly.

Some details are used without defining terms appropriately.

In order for me to improve my rating, I would like the questions / concerns I had raised addressed. In particular, I am worried that some conclusions are drawn without appropriate control over the experiments.

---

> ### Author Response · Authors · 2022-11-13
> **Response to JaMR**
>
> Dear reviewer,
>
> We thank the reviewer for their detailed review. It seems like the reviewer's main suggestion was to provide more evidence to support the claim that an aligned objective is responsible for the strong performance of ALM, relative to SAC-SVG. We have incorporated this suggestion by running two new ablation experiments:
> 1) In figure 15 (Pg 28), we compare ALM with an ablation where the model is leant to maximize likelihood and encoder to minimize reconstruction error. ALM significantly outperforms this variant on all 5 tasks, strengthening the claim about the aligned objective being responsible for the strong performance.
> 2) To test whether the soft actor critic’s entropy used in SAC-SVG can be a confounding factor causing SAC-SVG to perform worse than ALM, we compare a version of ALM using a soft actor critic entropy bonus like the SAC-SVG. In our new Table 4 (Pg 27), on two difficult tasks, this variant outperforms SAC-SVG, even outperforming ALM in one of them.
>
> **Do these new experiments, together with the answers and revisions described below, address all the reviewer's concerns?**
>
> > could you write how a sentence about how the methods mentioned (decision-aware loss functions, etc) tackle the objective mismatch problem?
>
> We have added a line in the 3rd para of the related work section, about prior decision aware loss functions describing that they optimize the model to minimize the difference between true and imagined next step values.
>
> > Defining the Q-function
>
> We have added the definition of true Q function at the end of the Preliminaries sub section (Pg 3). The same definition is used for the Q function in all the proofs. We denote the learnt Q function, parameterised by $\theta$ using $Q_{\theta} (s_t, a_t)$.
>
> > details about the classifier used in Section 4 should be moved to the main text.
>
> We have done this.
>
> > I am confused about the the second line of Equation 4. How is this infinite product carried out in practice? From Algorithm 1, it seems only the first K terms are used. If so, why is this part in Equation 5?
>
> The terms inside our objective (Equation 5), only depend on the first K terms of the learnt model-based distribution q(\tau).  We see the scope of confusion in the way equation 4 was written and we have changed it such that q^{K}(\tau) only includes the first K rollouts. This is why our method (Algorithm 1) only requires rolling out the latent-space model q^{K}(\tau) for K steps.
>
>
> > In MBRL, I thought that the dynamics model is used to sample data that can be used for training the policy. Does this method do this? I might be missing something, but from Algorithm 1, I can only see that samples are taken from the environment?
>
> Our method does sample data from the dynamics model for training the policy. This is done implicitly on Line 8 of Algorithm 1: the lower bound depends on samples from the dynamics model. We have added a modified line 8 of algorithm 1 to clarify that this step entails sampling from the learned dynamics model.
>
>
> > Does the Soft-actor critic nature of the main baseline affect any of the conclusions? are there any other differences between the methods [ours and SAC-SVG] that could lead to the differences in performance?
>
> Where possible, we have tried to use the same hyperparameters and architectures as SAC-SVG. We use the same training hyperparameters like update to data collection ratio, batch size and rollout length when compared to sac-svg. Both methods use the same policy improvement technique: stochastic value gradients.
>
> The two exceptions are decision decisions that simplify our method: unlike SAC-SVG, we use a feedforward dynamics model instead of an RNN; unlike SAC-SVG, we use a simple random noise instead of a more complex entropy schedule for exploration. Our new ablation in Table 4 (Pg 27), shows that our simpler exploration schedule is not responsible for the benefits of ALM, but training the representations and the model using the same objective as the policy is.

---

> > ### Author Response · Authors · 2022-11-16
> > **Response to JaMR**
> >
> > Dear reviewer,
> >
> > We hope that you've had a chance to read our response. **Does this clarify any of the concerns?** If not, please let us know and we can further revise the paper or clarify any concerns. We look forward to continuing the discussion.

---

### Official Review · Reviewer_RXsd · 2022-11-01

**Confidence:** 3
**Correctness:** 3
**Technical Novelty And Significance:** 3
**Empirical Novelty And Significance:** 3
**Recommendation:** 6

**Clarity, Quality, Novelty And Reproducibility:**

This paper is well-written and of good quality in terms of empirical evaluation. Following previous work (Eysenbach et al., 2021) that proposes a single objective for the policy and the state-space model, it derives a different objective for the policy and the latent-space model. The novelty of this paper has two aspects: (1) Different from the previous objective that uses a state-space model, it uses a latent-space model. (2) The objective derived in this paper covers the scenario with k-step rollouts and is a generalization of the previous objective. However, it should be noted that while the theory is nice and has some novelty, the practical algorithm deviates from it.

On reproducibility, the source code is included in the supplementary material.

**Strength And Weaknesses:**

The strength of the paper is that the proposed practical algorithm achieves superior performance on some continuous control tasks without using the common ensemble technique for MBRL. In addition, an abundance of ablation experiments is performed to validate parameter K's sensitivity, the components' effectiveness, and the learned values and representations.

However, there is a major weakness in this paper. The algorithm is inconsistent with the theory. Notably, the log of rewards and Q function in Eq (5) is omitted in the actual implementation. In addition, an extra hyperparameter is added to tune the effect of the KL term. More importantly, there is no ablation study on these two important design changes.

**Summary Of The Paper:**

This paper proposes a proxy model-based RL objective which is proved to be the lower bound of the usual RL objective. The proxy objective can be optimized with respect to the policy, the latent-space model, and the representation. Experiments on a range of continuous control tasks show that a practical algorithm loosely based on the proxy objective achieves better sample complexity than several competitive existing model-based and model-free algorithms.

**Summary Of The Review:**

Overall, this is a nice and well-written paper. It develops an extension of an existing objective (Eysenbach et al., 2021). It has strong empirical results for the practical algorithm it proposes. However, in my opinion, there is a major weakness - the practical algorithm deviates from the theoretical analysis without providing enough empirical/theoretical analysis.

Some questions to be addressed that impact the score:
1. What empirical evidence motivates using a latent-space model instead of a state-space model? How does it compare to MnM (Eysenbach et al., 2021)?
2. What empirical/theoretical evidence motivates omitting the log of rewards and Q function? Could you provide further justification for it?
3. How does the coefficient of the KL term affect performance?

Some suggestions for improvement that do not impact the score:
1. The definition of the Q function is in the appendix. It should be introduced in the main text before using it in Eq (5).
2. In Eq (7), I believe the KL term is reversed. It should be written as $KL(m_\phi(\cdot | z_i, a_i) || e_{\phi_\text{targ}}(\cdot | s_{i+1}))$. Otherwise, it is inconsistent with Eq (6) and the discussion in Sec 3.4.
3. Inaccurate/misleading statement after Eq (8): “we reduce the variance of this objective by computing this objective for multiple horizons and then taking an average.” However, in the experiment section, only fixed horizons are used.
4. Typo in the paragraph before the conclusion section: “we that”.
5. Typo in Lemma A.2: $Q(s_t, a_t)$ should be $Q(s_K, a_K)$.
6. In the proof of Theorem 3.1, I suggest changing ${d\tau dH}$ to ${dH d\tau}$ in equation g for better clarity.
7. In the proof of Theorem 3.1, it should be $H-1$ instead of $\max(0, H-1)$ above the summation. The latter will incorrectly calculate $\log e_\phi(z_1 | s_1) - \log m_\phi(z_1 | z_0, a_0)$ twice.
8. Appendix B, the comparison to prior work on lower bounds for RL (MNM), should be moved to the main text since it is the most relevant related work.
9. It would be nice if there were an analysis of the gap of the lower bound with respect to the model error.

Benjamin Eysenbach, Alexander Khazatsky, Sergey Levine, and Ruslan Salakhutdinov. Mismatched no more: Joint model-policy optimization for model-based rl, 2021. URL https://arxiv.org/abs/2110.02758.

---

> ### Author Response · Authors · 2022-11-13
> **Response to RXsd**
>
> Dear reviewer,
>
> We thank the reviewer for their detailed feedback. It seems like the reviewer's main suggestion is to add additional evidence to explain why the practical method deviates from the theoretical derivation, which we have incorporated by running 3 new experiments:
>
>
> 1) In figure 11 (Pg 26), we ablate the KL coefficient used for ALM, using values of 0.0, 0.05, 0.15, 0.1, 0.2, 0.5, 1. ALM performs similarly for all values, indicating that our method is not sensitive to this hyperparameter.
>
> 2) In figure 12 (Pg 26), we show that adding the log of rewards / Q function hurts the performance of ALM, thus motivating its omission. However, in Appendix D1 we show how this theoretically-motivated log term can be included while retaining high-performance.
>
> 3) In figure 13 (Pg 27), we compare ALM (a latent-space method) to an ablation of our method designed to resemble MnM (a state-space method). ALM outperforms this ablation, supporting our claim that learning a latent-space model is preferable to a state-space model.
>
>  **Together with answers to specific questions, does this address all the reviewer's concerns?**
>
> > Some suggestions for improvement that do not impact the score:
>
> Thanks for these suggestions. We have incorporated them into the paper, using red text to indicate the changes. We welcome any additional suggestions for improving the paper.

---

> > ### Comment · Reviewer_RXsd · 2022-11-15
> > **Response to Authors**
> >
> > Dear authors,
> >
> > Thank you for the additional experiments. They are interesting and have answered some of my questions. However, I still have a few concerns.
> >
> > Firstly, it looks like the performance when $c=0$ (the coefficient for the KL term) is quite similar to the performance when $c$ takes other values (Fig. 11). It would be better if you could plot a sensitivity curve to show that the intermediate value of $c$ achieves better performance especially compared to $c=0$. The sensitivity curve could be one in which the x-axis is the value of $c$, and the y-axis is a performance metric (the area under the training curve, etc.)  This, together with Fig. 5a (no classifier), will help support the motivation that the policy should take the model into account.
> >
> > Otherwise, since the value loss doesn’t help much in model learning (also a concern of reviewer *PJmH*, see Fig. 5a no value, Fig. 12 Semi log rewards), the theory would be completely vacuous. In other words, the “interactions” derived from the lower bound between the model and the policy were not helping.
> >
> > Secondly, thank you for pointing out this interesting relationship between the log and linear functions. However, I am still confused about the “Transformed log reward” experiment. I’ve also read your relevant response to reviewer *km6q*. Specifically, I have two questions: 1) what is the actual loss you use for “Transformed log reward”? I fail to see how the consistency between the “Transformed log reward” and the theory and the approximate equivalence between the “Transformed log reward” and the original implementation can be satisfied simultaneously based on the given details. 2) For why using the log rewards fails, the caption under Fig. 12 mentions a hypothesis that the non-linearity of log(x) makes the Q-values similar for different actions. Wouldn’t add a large constant to the rewards make the absolute difference between the Q-values smaller?
> >
> > Also, as reviewer *PJmH* points out, the KL term in Eq (7) is not updated. Is the KL term in the paper defined as $KL( e_{\phi_\text{targ}}(\cdot | s_{i+1}) || m_\phi(\cdot | z_i, a_i) ) = \log{\frac{m_\phi(\cdot | z_i, a_i)}{ e_{\phi_\text{targ}}(\cdot | s_{i+1}) }}$? If so, I suggest changing it to be consistent with the standard notation or clearly defining it.

---

### Decision · Program_Chairs · 2023-01-20

**Decision:**

Accept: poster

**Justification For Why Not Higher Score:**

A considerable gap between theory and algorithm makes the work somewhat weak.


**Justification For Why Not Lower Score:**

The paper still brings interesting directions for model-based RL.


**Metareview: Summary, Strengths And Weaknesses:**

This work proposes a single objective for learning the representation, the model, and the policy, which is shown to be a lower bound on the expected returns. The reviewers are generally appreciative of the work. Developing the objective and the consequent algorithms is an important step in model-based RL. Hence, I recommend accepting this paper.

I strongly recommend that the authors update the draft to incorporate the reviewers' suggestions. Most importantly, the characterization of the gap between theory and algorithm should be acknowledged and described in the paper. It should be done carefully by adjusting some of the inconsistent claims in the paper, such as the following:

- “Our method jointly optimizes all three components using a single objective, which is a lower bound on expected returns.” (Page 1, Figure 1).
- “Compute lower bound using off-policy actions,” “Update encoder and model by gradient ascent on off-policy lower bound,” etc. (Page 5, Algorithm 1).
- “the actions maximizing the lower bound” (Page 8, Sec. 5.3).
- “results in a method where the representations, model, and policy all ‘cooperate’ to maximize a lower bound on the expected returns” (Page 9, Sec. 6).

These claims cannot hold, as the authors acknowledged that the objective the algorithm is optimizing is not guaranteed to be the lower bound.

Hence, acceptance should be conditional on these changes.

Also, please note the following suggestions.
In the updated draft, the added definition of the action-value function should have $s_t$ and $a_t$ given as conditions in the expectation. In the subscript of the expectation, $a_t$ shouldn’t appear as a random variable drawn from $\pi$.
It is perhaps not the best choice that Section 3.3 ends with the new objective without giving any intuition. The intuition appears in the “connections with prior work” section, which is perhaps not the best section to look for intuition, either. At least it took me a while to realize that the intuition appears there.


**Note From Pc:**

if the above contains the word "oral" or "spotlight" please see: "oral" presentation means -> notable-top-5% and "spotlight" means -> notable-top-25%. As stated in our emails, we are disassociating presentation type from AC recommendations